



# The NO$_2$ camera based on Gas Correlation Spectroscopy

Leon Kuhn[1, 2], Jonas Kuhn[1, 2], Thomas Wagner[1, 2], and Ulrich Platt[1, 2]

[1]Institute for Environmental Physics, University of Heidelberg, Germany
[2]Max-Planck Institute for Chemistry, Mainz, Germany

**Correspondence:** Leon Kuhn (l.kuhn@mpic.de)

**Abstract.** Monitoring of NO$_2$ is in the interest of public health, because NO$_2$ contributes to the decline of air quality in many urban regions. Its abundance can be a direct cause of asthmatic and cardiovascular diseases and plays a significant part in forming other pollutants such as ozone or particulate matter. Spectroscopic methods have proven to be reliable and of high selectivity by utilizing the characteristic spectral absorption signature of trace gasses such as NO$_2$. However, they typically

lack the spatio-temporal resolution required for real-time imaging measurements of NO$_2$ emissions. We propose imaging measurements of NO$_2$ in the visible spectral range using a novel instrument, an NO$_2$ camera based on the principle of Gas Correlation Spectroscopy (GCS). For this purpose two gas cells (cuvettes) are placed in front of two camera modules. One gas cell is empty, while the other is filled with a high concentration of the target gas. The filled gas cell operates as a non-dispersive spectral filter to the incoming light, maintaining the two-dimensional imaging capability of the sensor arrays. NO$_2$ images are

generated on the basis of the signal ratio between the two images in the spectral window between 430 and 445 nm, where the NO$_2$ absorption cross section is strongly structured. The capabilities and limits of the instrument are investigated in a numerical forward model. The predictions of this model are verified in a proof-of-concept measurement, in which the column densities in specially prepared reference cells were measured with the NO$_2$ camera and a conventional DOAS instrument. Finally, results from measurements at a large power plant, the Großkraftwerk Mannheim (GKM), are presented. NO$_2$ column densities of

the plume emitted from a GKM chimney are quantified at a spatio-temporal resolution of 1/12 frames per second (FPS) and 0.92 m × 0.92 m. A detection limit of $1.89 \cdot 10^{16}$ molec cm$^{-2}$ was reached. An NO$_2$ mass flux of $F_m = (7.41 \pm 4.23)$ kg h$^{-1}$ was estimated on the basis of momentary wind speeds obtained from consecutive images. The camera results are verified by comparison to NO$_2$ slant column densities obtained from elevation scans with a MAX-DOAS instrument. The instrument prototype is highly portable and cost-efficient at building costs of below 2,000 Euro.

## 20  1  Introduction

Oxides of Nitrogen (NO$_x$ = NO + NO$_2$) play an important role in urban air quality. Nitrogen dioxide (NO$_2$) is itself toxic to humans and furthermore contributes to the formation of ozone (O$_3$) and particulate matter. Both NO$_2$ as well as ozone and particulate matter are linked to a variety of diseases, such as asthmatic and cardiovascular diseases. It is estimated, that 7-8 % of all European citizens are exposed to an annual mean exceeding 40 $\mu$g m$^{-3}$, which is the exposure limit recommended by

the World Health Organization (WHO, 2000). In other parts of the world exceedances are even higher. Therefore, monitoring NO$_2$ emissions and abundance near the planetary surface is of interest. In many cases the NO$_2$ concentration gradients of





interest occur on small spatial (sub-meter) and temporal (sub-second) scales, e.g. when measuring the emissions of moving point sources, such as cars, ships, or air planes. At the same time examinations of plume geometries, mass fluxes, and chemical reactions that take place in plumes require spatial coverage of the scene. Overall, an imaging method for $NO_2$ with high

spatio-temporal resolution could reveal more insight into the quantity and the dynamics of $NO_2$ emissions.

In polluted regions $NO_x$ emissions are mainly of anthropogenic origin. Combustion processes, which occur e.g. in car motors or industrial power plants, generate $NO_x$, which, at the time of emission, consists mostly of NO (typically with $NO_2/NO_x$ ratios as low as 5-10 %, see e.g. Kenty et al. (2007); Carslaw (2005)). Through oxidization processes, such as

$$NO + O_3 \rightarrow NO_2 + O_2 \tag{R1}$$

or, at very high NO concentrations,

$$2NO + O_2 \rightarrow 2NO_2 \tag{R2}$$

NO is converted to $NO_2$. Besides, other sources of $NO_x$ exist, such as geophysical events like lightning strikes, forest fires or soil emissions. Due to photodissociation, i.e.

$$NO_2 + h\nu \rightarrow NO + O \tag{R3}$$

an equilibrium between $NO_2$, NO, and $O_3$ (quickly formed by $O + O_2$), called the Leighton relationship, settles in.

There are different remote sensing methods for monitoring of atmospheric trace gasses such as $NO_2$. The state of the art method is Differential Optical Absorption Spectroscopy (DOAS, Platt and Stutz (2008)), where the absorption cross sections of the target gasses are fitted to the spectrally resolved differential optical depths along a light path. Then the column densities of the target gasses are retrieved as fit parameters. DOAS measurements can be based on either natural light sources, such

as scattered sunlight, or on artificial ones. Modern DOAS spectrographs typically have a spectral resolution of $< 1$ nm and operate in the UV and visible spectral range. The benefits of analysing spectrally resolved data are high selectivity and low detection limits. However, grating spectrographs are less suited for imaging, because spectral mapping leads to a reduced light throughput. Therefore measurements with sufficient spatial and spectral resolution require rather long acquisition times of many minutes (Bobrowski et al. (2006); Louban et al. (2009)). Imaging DOAS (I-DOAS) is typically realized using a push-

broom technique, where one detector dimension is used for spatial resolution and the other for spectral mapping. Consequently I-DOAS requires to scan a field of view (FOV) column by column or row by row. This strategy was used, for example, by Manago et al. (2018), who report on an imaging DOAS instrument for $NO_2$, based on a hyperspectral camera with a spatial resolution of $640 \times 480$ pixels, a $13° \times 9°$ FOV and a frame rate of 0.2 FPS. Although modern hyperspectral cameras can reach adequate spatio-temporal resolution, problems like the immanent asynchrony of the push-broom scheme, as well as portability

and price of the instrumental setup remain.

We propose an imaging instrument for $NO_2$ based on Gas Correlation Spectroscopy (GCS, see e.g. Ward and Zwick (1975); Drummond et al. (1995); Wu et al. (2018)) and demonstrate that an instrument designed to measure only a single trace gas can work by using reduced but specific spectral information in order to maximize spatio-temporal resolution. This is achieved





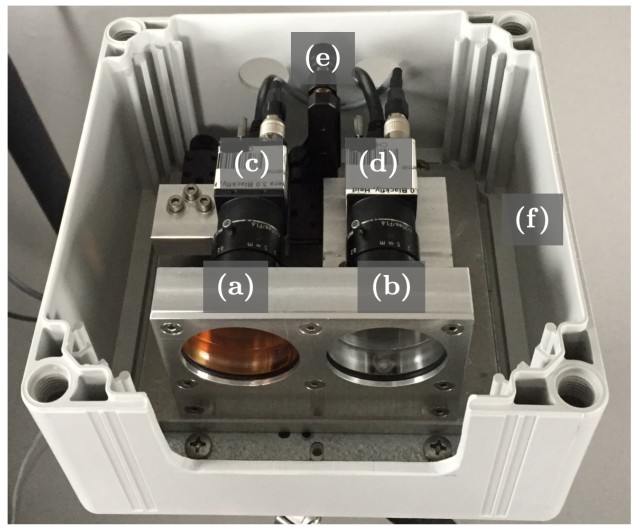

**Figure 1.** Photograph of the GCS-based $NO_2$ camera. The main parts of the instrument (see also Fig. 2 (a) and (b)) are two gas cells, one empty **(a)** and one filled with $NO_2$ **(b)**, as well as two camera modules **(c)**, **(d)**, each with a lens and a bandpass filter. One of the camera modules is placed on a mounting stage **(e)** which allows for precise alignment of the optical axes. All parts are mounted into a plastic case **(f)**.

by the use of two 2D-photosensors, each equipped with a lens and a glass cell: one filled with air (the "empty" cell), and one filled with a high concentration of $NO_2$. Figure 1 shows a photograph of an instrument prototype. The $NO_2$ cell functions as a spectral filter to the incoming light, while the empty cell has ideally no effect on the incoming light and serves as a reference. At the same time, the cameras fully resolve the light in two spatial dimensions. This way we obtain image data with only two spectral channels (in contrast to about 100 spectral channels used for typical DOAS fitting windows). The $NO_2$ column density measured by each pixel of the instrument can then be obtained by application of the Lamber-Beer law to the two channels. This principle is explained in more detail in sect. 2.1. The method is therefore similar to the recently developed filter correlation based $SO_2$ camera (Mori and Burton (2006)), the imaging Fabry-Perot interferometer correlation spectroscopy technique (IFPICS, see e.g. Kuhn et al. (2019); Fuchs et al. (2021)) or the acousto-optical tunable filter (AOTF) based $NO_2$ camera (Dekemper et al. (2016)). However, using a gas cell has substantial advantages compared to the listed techniques. While the filter correlation approach through its reduced selectivity only works for large volcanic $SO_2$ emissions, Fabry-Perot interferometers and AOTFs require collimated light beams within the lens setup, largely reducing the light throughput. In order to further increase selectivity to $NO_2$, we use an additional bandpass filter with transmission in the region of 425 nm to 450 nm, where the absorption cross section of $NO_2$ shows strong characteristic features. An instrument of this kind requires that $NO_2$ can be stably contained in glass cells. The instrument prototype we present fulfils this requirement. The chemistry of $NO_2$ gas cells is explained in detail by Platt and Kuhn (2019).



The rest of this paper is structured as follows: Section 2 deals with the theory of GCS and how it can be utilized for imaging measurements of $NO_2$. We introduce an instrument forward model, which allows to predict instrument responses, detection limits, and cross sensitivities of a GCS-based $NO_2$ camera under different circumstances. Section 3 presents a prototype of the instrument and lists its detailed technical specifications. Section 4 shows the results of two measurements that have been taken with that instrument prototype. The first is a proof-of-concept measurement with reference cells in an optical laboratory. The

purpose of this measurement is to verify the functionality of the instrument and to validate the predictions of the instrument forward model in sect. 2.2. The second is a measurement of the emissions of the German coal power plant Großkraftwerk Mannheim (GKM). Section 5 concludes.

## 2 Theory

### 2.1 Gas Correlation Spectroscopy

The absorption of light is described by the Lambert-Beer law. It states that for a given incident spectral radiance $L_0(\lambda)$ the spectral radiance $L(\lambda)$ after travelling along a light path $s$ through absorbing media with absorption cross sections $\sigma_k(\lambda)$ and concentrations $c_k$ is given by:

$$L(\lambda) = L_0(\lambda) \cdot e^{-\sum_k \sigma_k(\lambda) \cdot \int c_k(s) \, ds} \tag{1}$$

$$= L_0(\lambda) \cdot e^{-\sum_k \sigma_k(\lambda) \cdot S_k} \tag{2}$$

$$= L_0(\lambda) \cdot e^{-\tau(\lambda)} \tag{3}$$

Here, $S_k = \int c_k(s) \, ds$ in units of $[\text{molec cm}^{-2}]$ denotes the column density of the absorbing medium $k$ in the atmosphere and $\tau$ is the resulting optical depth. In our application, $L_0$ denotes the radiance spectrum of scattered sunlight. The Lambert-Beer law can be applied to radiances, denoted with $L$ in units $[\text{W nm}^{-1} \text{ m}^{-2} \text{ sr}^{-1}]$, as well as to irradiances, denoted with $I$ in units of $[\text{W nm}^{-1} \text{ m}^{-2}]$.

The pixels of a photosensor do not resolve spectrally. Let $\mu_p(\lambda)$ be the number of photons per wavelength interval and time period in units of $[\text{ph nm}^{-1} \text{ s}^{-1}]$, that a photosensor is exposed to. It will then measure a detector signal $J$ in units of photoelectrons ([phe]), given by the spectral and temporal integral

$$J = \int_0^{t_{\text{exp}}} \int_0^\infty \eta(\lambda) \cdot \mu_p(\lambda) \, d\lambda \, dt \tag{4}$$

where $\eta$ in units of $[\text{phe ph}^{-1}]$ denotes the quantum efficiency of the photosensor. The wavelength dependence of $\eta$ typically

restricts the integration to the near ultra violet (UV), the visible, and near infrared regions of the electromagnetic spectrum. $\mu_p(\lambda)$ can be expressed as

$$\mu_p(\lambda) = L_0(\lambda) \cdot T(\lambda) \cdot E \cdot \frac{\lambda}{hc} \cdot e^{-\tau(\lambda)} \tag{5}$$

$$:= \tilde{\mu}_p(\lambda) \cdot e^{-\tau(\lambda)} \tag{6}$$



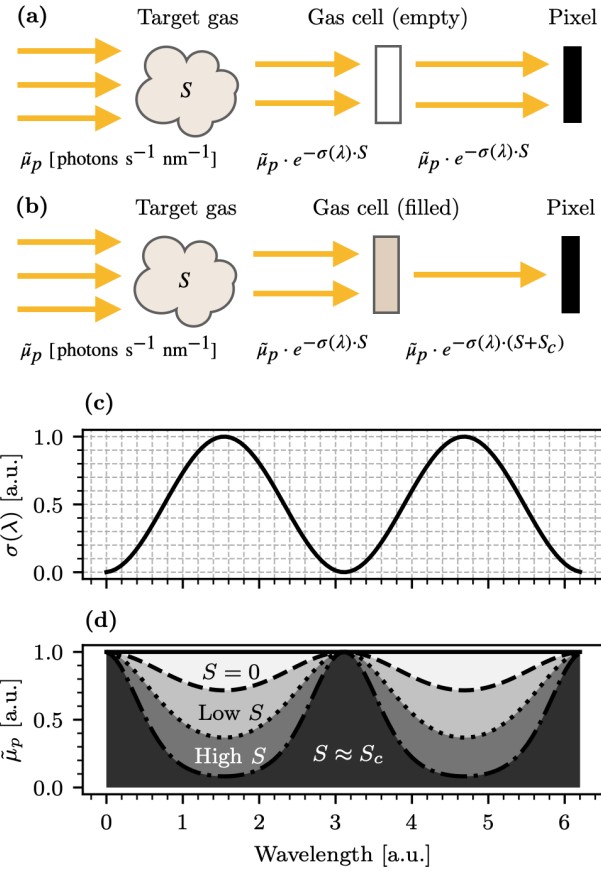

**Figure 2.** Schematic depiction of the absorption of incoming light in a GCS-based instrument. For simplicity only a single absorber is assumed. **(a)** shows the absorption scheme for the channel with the empty cell and **(b)** shows the absorption scheme for the channel with the filled cell. $S$ denotes the column density of the target gas and $S_c$ the column density of the target gas in the gas cell of the instrument. **(c)** and **(d)** demonstrate the principle of GCS: Given a hypothetical absorption cross section (here assumed to be of sinoidal shape, displayed in (c)), the spectral absorption can be derived from the Lambert-Beer law for different choices of $S$ (here $S = 0$, "Low $S$", "High $S$"). A photosensor is only sensitive to the spectrally integrated radiance that it is exposed to, i.e. the gray-coloured areas displayed in **(d)**.

where $L_0$ denotes the radiance spectrum of the light source, $T$ denotes the transmission of the instrumental setup, $e^{-\tau(\lambda)}$
describes all absorption along the light path according to the Lambert-Beer law, and $E$ denotes the étendue of the instrument in units of $[\mathrm{mm^2\ sr}]$. The factor $\lambda/hc$ converts radiant flux in units of [W] to photon counts per time, i.e. $[\mathrm{ph\ s^{-1}}]$, where $\lambda$ denotes wavelength, and $hc = 1.986 \cdot 10^{-25}$ J m denotes the product of Planck's constant and the speed of light.

Figure 2 explains the principle of GCS, assuming (for the sake of simplicity) that the target gas with column density $S$ and absorption cross section $\sigma(\lambda)$ is the sole absorber and thus $\tau = \sigma(\lambda) \cdot S$. Two camera modules are placed behind two gas cells,
of which one is filled with air (the "empty" cell) and one is filled with a high concentration of the target gas (see Fig. 1). For a





detector pixel with the indices $(i, j)$, the camera with the empty cell will measure

$$J_{(i,j)} = \int_0^{t_{\text{exp}}} \int_0^\infty \eta(\lambda) \cdot \tilde{\mu}_p \cdot e^{-\sigma(\lambda) \cdot S_{(i,j)}} \, \mathrm{d}\lambda \, \mathrm{d}t \tag{7}$$

and the camera with the cell containing the target gas will measure

$$J_{c,(i,j)} = \int_0^{t_{\text{exp}}} \int_0^\infty \eta(\lambda) \cdot \tilde{\mu}_p \cdot e^{-\sigma(\lambda) \cdot (S_{(i,j)} + S_c)} \, \mathrm{d}\lambda \, \mathrm{d}t \tag{8}$$

where $S_{(i,j)}$ denotes the column density of the target gas in the FOV of the pixel with indices $(i, j)$ and $S_c$ the column density of the target gas in the gas cell of the instrument. In imaging GCS, the instrument response (instrument signal)

$$\tilde{\tau}_{(i,j)} = \ln\left(J_{c,(i,j)}/J_{(i,j)}\right) \tag{9}$$

is computed for each individual pixel. $\tilde{\tau}_{(i,j)}$ is the logarithmic signal ratio between the two spectral channels of the instrument and functions as a measure of $S_{(i,j)}$: When $S_{(i,j)}$ is small, incoming light will be only slightly attenuated before it reaches

the cells, and thus the signal ratio $J_{c,(i,j)}/J_{(i,j)}$ will be smaller compared to a scenario in which $S_{(i,j)}$ is large and thus the atmospheric target gas has already attenuated a larger portion of the light, that else would have been absorbed by the gas cell. It therefore follows directly that $\tilde{\tau}_{(i,j)}$ grows monotonically with $S_{(i,j)}$.

     When using two camera modules with distinct optical setups, the resulting detector signals are highly sensitive to imperfections in the optical path. For example, small differences in the focal lengths of the camera lenses or dust particles on the lenses

or gas cells can induce significant false signals, contributing to $\tilde{\tau}$. Furthermore, vignetting is immanent to imaging measurements and manifests itself in increasing false signal gradients towards the corners of the image. These effects can be partly corrected by recording reference signals $J_{\text{ref},(i,j)}$ for the channel with the empty cell and $J_{c,\text{ref},(i,j)}$ for the channel with the filled cell in zenith direction, where $S = 0$ is assumed. In reality this latter condition need not be perfectly fulfilled, although it is important that $S$ is approximately constant throughout the FOV for the reference images. In analogy to eq. (7) and (8) the

reference signals are given by

$$J_{\text{ref},(i,j)} = \int_0^{t_{\text{exp}}} \int_0^\infty \eta(\lambda) \cdot \tilde{\mu}_p \, \mathrm{d}\lambda \, \mathrm{d}t \tag{10}$$

and

$$J_{c,\text{ref},(i,j)} = \int_0^{t_{\text{exp}}} \int_0^\infty \eta(\lambda) \cdot \tilde{\mu}_p \cdot e^{-\sigma(\lambda) \cdot S_c} \, \mathrm{d}\lambda \, \mathrm{d}t \tag{11}$$

The measurement signal ratio is then divided by the reference signal ratio, i.e.

$$\tilde{\tau}_{(i,j)} = \ln\left( \frac{J_{c,(i,j)} \cdot J_{\text{ref},(i,j)}}{J_{(i,j)} \cdot J_{c,\text{ref},(i,j)}} \right) \tag{12}$$

This procedure is also referred to as flat field correction. In the following section it will be shown that in good approximation $\tilde{\tau} \propto S$ holds.



## 2.2 Instrument model calculation

A numerical forward model was implemented to predict characteristics of a GCS-based $NO_2$ camera. Specifically we investi-
gate the shape of the instrument response, the calibration curve, and the signal-to-noise ratio (SNR) as a function of $S$ and $S_c$,
as well as cross sensitivities to other atmospheric trace gasses. This section discusses specifically the application of GCS to
measurements of $NO_2$. Other trace gasses may, for example, require to operate in a different spectral range. Overall, the sim-
ulation of realistic conditions of daytime measurements in the atmosphere is the aim. For this a spectrum of scattered sunlight
is used as the light source and atmospheric $NO_2$ column densities are considered in the range from $10^{16}$ to $10^{18}$ molec cm$^{-2}$,
as well as integration times on the scale of seconds. The relevant detector signals are modelled according to eq. (7), (8), (10),
and (11). In this instrument model, we assume $T(\lambda) = T_f(\lambda) \cdot T_l(\lambda)$, where $T_f$ denotes the transmission of the bandpass filter
and $T_l$ denotes the transmission of the camera lens. Since $t_{\exp}$ is realistically small enough that $I_0(\lambda)$ is constant throughout
exposure and the transmission of the bandpass filter used is effectively a cut-off function outside its transmission band from
430 nm to 445 nm, the detector signals can be simplified to

$$J(\tau) = t_{\exp} \cdot \int_{430 \text{ nm}}^{445 \text{ nm}} \eta(\lambda) \cdot \tilde{\mu}_p(\lambda) \cdot e^{-\tau} \, \mathrm{d}\lambda \tag{13}$$

The choice of of this particular bandpass filter is motivated by the strong, characteristic absorption features, that $NO_2$ shows in
its transmission range. The absorption cross section of $NO_2$ (Vandaele et al. (2002)) is displayed in Fig. 3 (a) with a zoomed-in
region close to the transmission band of the bandpass filter. The model requires a light source radiance spectrum $L_0$. For realistic
applications of the instrument the light source will almost exclusively be an atmospheric background spectrum, i.e. a radiance
spectrum of scattered sunlight. We use a highly resolved irradiance spectrum in units of [W nm$^{-1}$ m$^{-2}$] (Chance and Kurucz
(2010)), and scale it with a low-resolution radiance spectrum at 400 nm (Pissulla et al. (2009)) in units of [W nm$^{-1}$ m$^{-2}$ sr$^{-1}$].
This way we obtain a radiance spectrum that represents the typical spectral shape of scattered sunlight, but maintain the high
spectral resolution of the irradiance spectrum. We argue that this is the most realistic general estimation of the background
spectrum that we can make. The radiance spectrum used for scaling was recorded at Thessaloniki, Greece, at a sun zenith angle
of 21°. The transmission lines of the bandpass filter $T_f(\lambda)$ and the camera lenses $T_l(\lambda)$, as well as the quantum efficiency
of the camera sensors $\eta(\lambda)$ are provided by the manufacturers. An étendue of $E \approx 10^{-5}$ sr mm$^2$ was assumed throughout,
which was computed on the basis of a fully opened aperture (f-number 1.6). Figure 3 (b) shows plots of $L_0$, $T_f$, $T_l$ and $\eta$. In
the following we assume an exposure time of 2 s throughout. The detector signals $J$, $J_c$, $J_{\text{ref}}$ and $J_{c,\text{ref}}$ are then calculated
by numeric integration, according to the instrument model as described. Figure 4 shows the modelled instrument response $\tilde{\tau}$
(see eq. (12)) as a function of the column density $S$ in the range from $10^{16}$ to $10^{18}$ molec cm$^{-2}$ for different choices of the
column density $S_c$ within the $NO_2$ cell of the instrument. The instrument response is in good approximation proportional to $S$.
The instrument calibration factor $k$ can be obtained for any fixed value of $S_c$ by sampling the instrument signal $\tilde{\tau}$ for different
choices of $S$ and fitting a linear function of the form

$$\tilde{\tau}(S) = k \cdot S \tag{14}$$





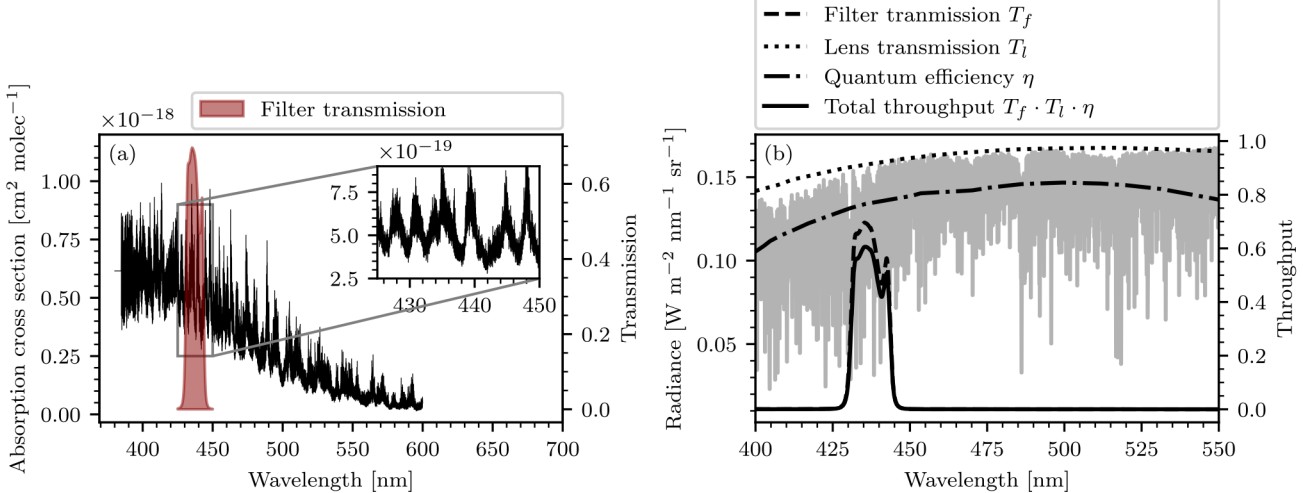

**Figure 3. (a)** The absorption cross section of $NO_2$. The red filled region marks the transmission region of the bandpass filter used in our instrument. The inset in the top right shows a zoomed-in view on a spectral range that contains the filter transmission and shows highly structured absorption features. **(b)** The radiance spectrum, as well as the transmission lines of the filter $T_f$ and the lens $T_l$, the quantum efficiency of the camera sensor $\eta$ and the total throughput $T_f \cdot T_l \cdot \eta$, that are assumed in the instrument model.

to the samples. In order to convert the unitless instrument signal $\tilde{\tau}$ to column densities, the inverse $k^{-1}$ in units [molec cm$^{-2}$] is used.

With this model we can also quantify the signal-to-noise ratio (SNR) in order to estimate the detection limit of the instrument under typical atmospheric conditions. An SNR of 1 is assumed to be the lower limit at which atmospheric column densities of the target gas can be resolved. Photoelectron counting follows Poissonian statistics, i.e. the uncertainty $\Delta J$ of a signal measured by a photosensor is $\Delta J = \sqrt{J}$. Thus, the uncertainty $\Delta \tilde{\tau}$ of the instrument signal $\Delta \tau$ can be expressed in closed form by application of Gaussian uncertainty propagation:

$$\Delta \tilde{\tau} = \frac{1}{\left( \sqrt{1/J + 1/J_c + 1/J_{\mathrm{ref}} + 1/J_{c,\mathrm{ref}}} \right)} \tag{15}$$

In practice the uncertainties of the reference signals will be comparably small, because the exposure time for the recording of $J_{\mathrm{ref}}$ and $J_{c,\mathrm{ref}}$ can be chosen to make the contribution of $1/J_{\mathrm{ref}}$ and $1/J_{c,\mathrm{ref}}$ negligible. Then the uncertainty reduces to

$$\Delta \tilde{\tau} = \frac{1}{\sqrt{(1/J + 1/J_c)}} \tag{16}$$

and the SNR can be expressed as

$$\mathrm{SNR} = \frac{\tilde{\tau}}{\Delta \tilde{\tau}} = \frac{\ln(J_c/J) - \ln(J_{c,\mathrm{ref}}/J_{\mathrm{ref}})}{\sqrt{(1/J + 1/J_c)}} \tag{17}$$





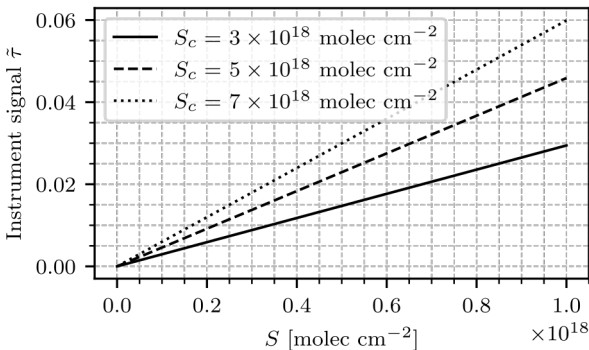

**Figure 4.** The modelled instrument signal $\tilde{\tau}_{(i,j)} = \ln\left(\dfrac{J_c \cdot J_{\mathrm{ref}}}{J \cdot J_{c,\mathrm{ref}}}\right)$ as a function of the target gas column density $S$ for different choices of cell column density $S_c$. The instrument response is almost perfectly linear in $S$. The slope of each line yields the instrument calibration corresponding to $S_c$.

This instrument model only accounts for the photon shot noise and disregards additional possible sources of noise such as dark noise and read-out noise of the photosensors. This is on purpose in order to make the model applicable to different instrumental

setups. In practice the shot noise is by far the dominating source of noise due to the large light throughput of the setup, and both dark current as well as dark noise can be neglected (see sect. 3 for a more detailed explanation). Figure 5 (a) shows the modelled SNR as a function of the cell column density $S_c$ for different choices of the column density of the target gas $S$. The highest SNR is reached at approximately $S_c \approx 4 \cdot 10^{18}$ molec cm$^{-2}$ with a slight dependence on the observed target gas column density $S$. Figure 5 (b) shows the modelled SNR as a function of the target gas column density $S$. The red horizontal

line marks the resulting detection limit, where SNR = 1. With an ideal choice of $S_c \approx 4 \cdot 10^{18}$ molec cm$^{-2}$, a detection limit of approximately $2 \cdot 10^{16}$ molec cm$^{-2}$ is reached with an exposure time of 2 s.

The instrument model also allows to study the selectivity of the instrument. Equation (13) holds under the assumption that the target gas is the sole absorber. In a realistic measuring scenario many different trace gasses other than NO$_2$ could be present in the atmosphere. Cross sensitivities to other trace gasses can be determined on the basis of the instrument model. We define

$\tau_X$, the false signal of a species X, as the additional contribution to the overall instrument signal $\tilde{\tau}$, that is due to the absorption of X, and present the results of a study on the false signals of water vapour (H$_2$O, absorption cross section was taken from Rothman et al. (2013)) and the oxygen collision complex (O$_4$, absorption cross section was taken from Thalman and Volkamer (2013)), since both species show possibly relevant absorption features in the spectral range our instrument operates in. Figure 6 shows the absorption cross sections of NO$_2$, H$_2$O, O$_4$, and the transmission line of the bandpass filter used. The bandpass filter

blocks almost all light of wavelengths greater than $\lambda \geq 445$ nm. Therefore most of the O$_4$ absorption is filtered out and $\tau_{O_4}$ is strongly reduced. Water vapour, on the other hand, shows strong absorption features between 440 and 445 nm. Calculating the false signals of the two species requires an assumption of their atmospheric abundance. In reality these column densities can vary strongly with place and time. We therefore use the model to make predictions on the cross sensitivities assuming





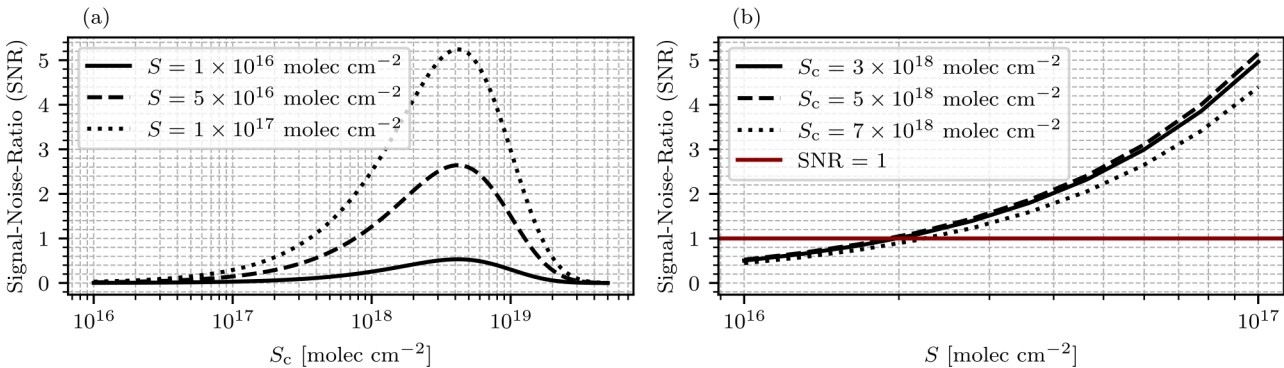

**Figure 5. (a)** Modelled SNR as a function of the cell column density $S_c$ for different choices of the target gas column density $S$. The highest SNR is reached for a cell column density of approximately $S_c \approx 4 \cdot 10^{18}$ molec cm$^{-2}$, with a slight dependence on $S$. **(b)** Modelled SNR as a function of the column density of the target gas $S$ for different choices of the cell gas column density $S_c$. The red vertical line marks SNR=1 and thus the detection limit of the instrument.

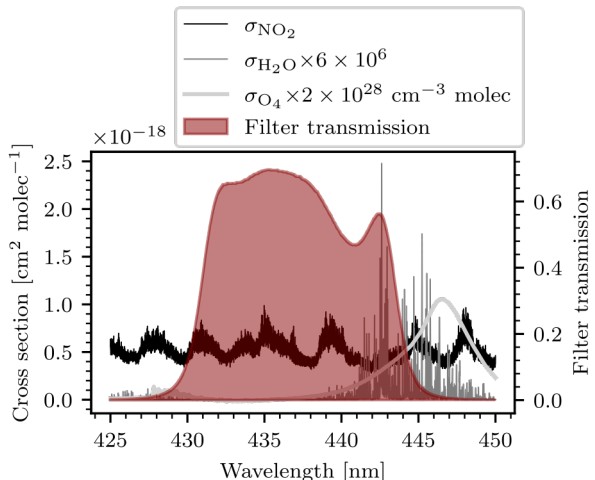

**Figure 6.** Cross sections of $NO_2$, $H_2O$, $O_4$ and the transmission of the bandpass filter (red shaded area) used. The cross sections of $H_2O$ and $O_4$ were scaled (see legend) in order to display them on a mutual axis.



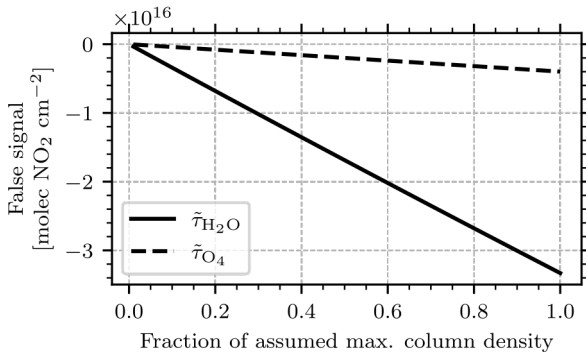

**Figure 7.** Modelled cross sensitivity to $H_2O$ and $O_4$. The ordinate shows the fraction of the assumed maximal column density for both species, which are $6 \cdot 10^{23}$ molec $cm^{-2}$ for $H_2O$ and $10^{44}$ molec$^2$ $cm^{-5}$ for $O_4$. The abscissa shows the false signal of the two species converted to $NO_2$ column density equivalents. The calibration of the model was obtained from Fig. 4, assuming a cell column density of $S_c = 4 \cdot 10^{18}$ molec $cm^{-2}$.

large, but still realistic column densities of the cross sensitive species. If the predicted false signals are sufficiently small,
the cross sensitivities can be neglected altogether, because the model has then realistically overestimated the induced false
signals. For $O_4$ a maximum column density of $10^{44}$ molec$^2$ $cm^{-5}$ at a light path length of 10 km was assumed. For reference,
Peters et al. (2019) report maximal $O_4$ column densities of around $5 \cdot 10^{43}$ molec$^2$ $cm^{-5}$ during the CINDI-2 measurement
campaign. For $H_2O$ a maximum column density of $6 \cdot 10^{23}$ molec $cm^{-2}$ was assumed. This corresponds to a relative humidity
of 100 % at a pressure of 1 atm, temperature of $20°$ C and a light path length of 10 km. Figure 7 shows the modelled false
signals of $H_2O$ and $O_4$. The false signal was converted to $NO_2$ column density equivalents using the calibration of the model
obtained from Fig. 4, assuming a cell column density of $S_c = 4 \cdot 10^{18}$ molec $cm^{-2}$. Both species induce a negative false signal.
When expressed in $NO_2$ signal equivalents, the false signal of $O_4$ is comparably small, reaching around $-2 \cdot 10^{15}$ molec $cm^{-2}$
assuming the maximal column. The false signal of $H_2O$ is an order of magnitude larger, reaching up to $-3.2 \cdot 10^{16}$ molec $cm^{-2}$
assuming the maximal column. As discussed, we treat these false signals and the column densities that have generated them
as an overestimate of a realistic expectation. In addition, the naturally abundant water vapour of the atmosphere is typically
distributed much more homogeneously than strong $NO_2$ concentration gradients from a point source. Under this circumstance
false signal induced by water vapour should be easily separable from the $NO_2$ signal of interest. Water vapour inside the plume
of a point source emission, which can not be separated from $NO_2$ signal by the argument above, is contained within much
shorter light paths (typically on the order of 100 - 200 m) and is not expected to induce relevant false signals.

## 2.3   Analytic instrument model

The instrument model presented in sect. 2.2 allows forward modelling of the measuring process with highly resolved radiance
spectra and absorption cross sections. However, the integral terms that occur in the instrument response do not allow for a





closed-form expression of $\tilde{\tau}$. Starting from eq. (12), we simplify the expression for the instrument response by assuming a constant radiance spectrum, $L_0(\lambda, t) = \text{const}$ and quantum efficiency $\eta(\lambda) = \text{const}$. We restrict the model to some spectral

range $\Delta\lambda = [\lambda_{\min}, \lambda_{\max}]$ and define $\lambda_{\mathrm{mid}} = (\lambda_{\max} + \lambda_{\min})/2$. The final assumption is that the cross section of the target gas consists of only two representative absorption strengths, $\sigma_{\mathrm{strong}}$ and $\sigma_{\mathrm{weak}}$. To determine both, we compute the median of $\sigma_{\mathrm{NO_2}}$ and define $\sigma_{\mathrm{weak}}$ and $\sigma_{\mathrm{strong}}$ as the mean absorption strength below and above the median respectively. The absorption cross section can then be expressed as

$$\sigma = \sigma_{\mathrm{weak}} \cdot \mathbf{1}_{[\lambda_{\min}, \lambda_{\mathrm{mid}}]} + \sigma_{\mathrm{strong}} \cdot \mathbf{1}_{[\lambda_{\mathrm{mid}}, \lambda_{\max}]} \tag{18}$$

where $\mathbf{1}_I$ is the indicator function on an interval $I$. The instrument response $\tilde{\tau}$ then only depends on the integrals of transmission terms $T_S := e^{-\sigma \cdot S}$ of the form

$$\int_{\Delta\lambda} T_S \, \mathrm{d}\lambda = \frac{\lambda_{\max} - \lambda_{\min}}{2} \cdot \left(e^{-\sigma_{\mathrm{weak}} \cdot S} + e^{-\sigma_{\mathrm{strong}} \cdot S}\right) \tag{19}$$

$$= \frac{\lambda_{\max} - \lambda_{\min}}{2} \cdot \left(T_{S,\mathrm{weak}} + T_{S,\mathrm{strong}}\right) \tag{20}$$

Equation (12) then takes the form

$$\tilde{\tau} = \ln\left(\frac{J_c \cdot J_{\mathrm{ref}}}{J \cdot J_{c,\mathrm{ref}}}\right) \tag{21}$$

$$= \ln\left(\frac{\int_{\Delta\lambda} T_S \cdot T_{S_c} \, \mathrm{d}\lambda}{\int_{\Delta\lambda} T_S \, \mathrm{d}\lambda \cdot \int_{\Delta\lambda} T_{S_c} \, \mathrm{d}\lambda}\right) \tag{22}$$

$$= \ln\left(\frac{2 \cdot (T_{S,\mathrm{weak}} \cdot T_{S_c,\mathrm{weak}} + T_{S,\mathrm{strong}} \cdot T_{S_c,\mathrm{strong}})}{(T_{S,\mathrm{weak}} + T_{S,\mathrm{strong}}) \cdot (T_{S_c,\mathrm{weak}} + T_{S_c,\mathrm{strong}})}\right) \tag{23}$$

This equation can be applied to arbitrary absorption cross sections, however $\sigma_{\mathrm{weak}}$ and $\sigma_{\mathrm{strong}}$ must be estimated anew for each absorption cross section. The analytical term in eq. (23) could be further simplified, if a gas without broadband contribution to

its absorption cross section were considered. In that case, $\sigma_{\mathrm{weak}} \approx 0$ and the column in the gas cell $S_c$ could be chosen, so that $T_{S_c,\mathrm{strong}} \approx 0$. The approximation of the instrument signal would then simplify to

$$\tilde{\tau} \approx \ln\left(\frac{2}{T_{S,\mathrm{strong}} + 1}\right) \tag{24}$$

The true instrument signal $\tilde{\tau}$, as obtained in sect. 2.1, and the analytical approximation in eq. (23) are plotted in Fig. 8. The spectral range of choice was 430 - 445 nm. The analytical approximation underestimates the true instrument response by

around 25%, but is equally linear in $S$ besides. The deviation can be corrected by tweaking the choice of $\sigma_{\mathrm{weak}}$ and $\sigma_{\mathrm{strong}}$, although good candidates can not be known a priori. The derived analytical expression allows for quick approximation of the sensitivity of a GCS measurement.

## 3 Instrument prototype

We have built an instrument prototype based on commercially available hardware. The camera modules use a monochrome

progressive scan CMOS sensor in a 1/1.2 " format with a pixel size of $5.86 \ \mu\mathrm{m} \times 5.86 \ \mu\mathrm{m}$ and a global shutter. They record





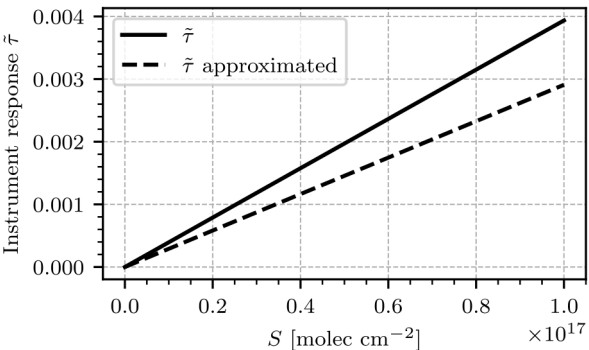

**Figure 8.** Comparison of the true instrument signal $\tilde{\tau}$, as obtained in sect. 2.1 (solid line), and the analytical approximation in eq. (23) (dotted line). A column density in the gas cell of $S_c = 4 \cdot 10^{18} \text{ molec cm}^{-2}$

images with $1920 \times 1200$ (height $\times$ width) pixels. A charge signal is digitized by a 16-Bit analog-digital converter (ADC). The cameras connect via USB 3.0 to a controlling computer equipped with corresponding camera software. Image acquisition rates depend on the selected exposure time and the read-out time $t_{\text{read}} = 24.39$ ms of the camera sensors. The instrument is therefore limited to a frame rate of 41 FPS at best. However, the read-out time $t_{\text{read}}$ can be reduced by using windowing, a

feature where the cameras are advised to only read out a subrange of their sensor arrays. The usability of windowing depends on the imaged scene and whether large parts of the FOV can be neglected. The camera modules have a read-out noise of 7 phe. The thermal dark signal of the camera modules was determined experimentally according to the EMVA (see Jähne, B. (2010)). A thermal dark signal of $(24 \pm 9)$ phe s$^{-1}$ at a sensor temperature of $50\,^{\circ}$C and a doubling temperature of $(6.1 \pm 0.1)\,^{\circ}$C were found. The camera modules have a full-well depth of 34,000 phe. Given that in bright daylight the exposure times for images

within the dynamic range of the camera are typically far below 1 s, the contribution of the dark signal to the total measured camera signal is negligibly small (e.g. below 0.05 % for an exposure time of 30 ms and a sensor saturation of 50 %). Also the total dark noise (meaning read-out noise + thermal noise) is negligible compared to the photon shot noise of around 130 phe at 50 % saturation. Each camera is equipped with a lens with a focal length of $f = 25$ mm. The full diagonal, vertical, and horizontal opening angles amount to $30\,^{\circ}$, $16.1\,^{\circ}$, and $25.5\,^{\circ}$, respectively. For each camera a bandpass filter with transmission

in the range from 430 - 445 nm was placed between the camera lens and the camera sensor. The gas cells of the instrument are cylindrical with a diameter of 50 mm and a thickness of 10 mm. The $NO_2$ cell was filled from a large reservoir to contain an $NO_2$ column density of $4 \cdot 10^{18}$ molec cm$^{-2}$ (which is the ideal value according to the results shown in sect. 2.2, specifically Fig. 5 (a)). The camera behind the $NO_2$ cell is mounted to a tiltable stage, which can be used to adjust its optical axis in vertical and horizontal orientation with mrad precision using two thumb screws. This adjustment is scene-dependent and of

crucial importance in order to eliminate shifts in the FOVs of the two cameras. All parts are placed inside a closable plastic case. Overall, the instrument is portable and compact, while maintaining a reasonable cost of below 2,000 Euro. A control software with graphical user interface was developed in the Python programming language.





## 4 Measurements

### 4.1 Proof-of-concept measurement with gas cells

In order to validate the instrument model described in sect. 2.2 we performed a simple laboratory experiment. Four glass cells were filled with different concentrations of $NO_2$ and measured with both the $NO_2$ camera and a conventional DOAS setup. The light source for the camera measurement was a halogen lamp inside an integrating sphere in front of which the cells were mounted onto a stand with a clamp. An additional series of images was recorded without a cell in the lightpath, whose average serves as the reference image ($J_{ref}$, $J_{c,ref}$, see eq. (12)). When evaluating the images taken by the $NO_2$ camera, an in-cell pixel
set and a background pixel set were defined. The in-cell pixel set contained the pixels inside the cell, while the background pixel set contained pixels of the illuminated entrance of the integrating sphere, not covered by the cell. Due to the varying size of the test cells, the in-cell and background pixel sets were different for each cell. The total acquisition time of the $NO_2$ camera was set to 3 minutes for each cell, and the exposure time of each camera was chosen such that the camera sensors saturated to approximately 50 %.

First, the column density inside the gas cell of the $NO_2$ camera was estimated as

$$S_c = \ln(\overline{J_{bg}}/\overline{J_{c,bg}})/\overline{\sigma} \tag{25}$$

where $\overline{J_{bg}}$ and $\overline{J_{c,bg}}$ are the camera signals of the camera with empty cell and the one with the filled cell respectively, averaged over the background pixels of all images. $\overline{\sigma} \approx 5.1 \cdot 10^{-19}$ cm$^2$ molec$^{-1}$ is the absorption cross section of $NO_2$, averaged over the spectral range from 430 to 445 nm. A cell column density of $S_c = (3.89 \pm 0.03) \cdot 10^{18}$ molec cm$^{-2}$ was obtained. The
cell was originally filled with $S_c = 4 \cdot 10^{18}$ molec cm$^{-2}$, but this deviation can be explained by the temperature-dependent $NO_2 \rightleftharpoons N_2O_4$ equilibrium. The lower the temperature, the lower the $NO_2$ concentration within the gas cell. The calibration of the instrument was obtained from the instrument model as explained in sect. 2.2. The fit procedure yielded a calibration factor of $k^{-1} = (2.69 \pm 0.02) \cdot 10^{19}$ molec cm$^{-2}$. Additionally, the signal offset $\tilde{\tau}_0$ of the instrument was calculated from the background pixels, which was defined as

$$\tilde{\tau}_0 = \ln(\overline{J_{c,bg}}/\overline{J_{bg}}) \tag{26}$$

Subtraction of $\tilde{\tau}_0$ from the instrument signal $\tilde{\tau}$ set the average background pixel to zero. The instrument signal of a test cell was determined by averaging over the pixels that were covered by the cell, i.e.

$$\tilde{\tau} = \ln(\overline{J_c}/\overline{J}) \tag{27}$$

where $J$ and $J_c$ denote the camera signal with the empty cell and with the filled gas cell respectively in the in-cell pixel region.
The uncertainty of these measurements is given by Gaussian error propagation according to eq. (16). The uncertainties $\Delta \overline{J_c}$ and $\Delta \overline{J}$ are obtained by computing the standard deviation of the detector signal in the in-cell region for the two channels respectively. Figure 9 (a) shows an exemplary image of this measurement. In the center foreground of the image the outline of test cell no. 4 and the stand and clamp, used to hold it, are shown. The offset $\tilde{\tau}_0$ was subtracted and the flat field correction was



**Table 1.** Column densities and instrument signal $\tilde{\tau}$ of each reference cell, measured with a DOAS instrument and the $NO_2$ camera.

| Cell no. | CD (DOAS) [molec cm$^{-2}$] | CD (camera) [molec cm$^{-2}$] | Instrument response $\tilde{\tau}$ | Model prediction for $\tilde{\tau}$ | Filter size |
|---|---|---|---|---|---|
| 1 | $(1.27 \pm 0.01) \cdot 10^{16}$ | $(0.99 \pm 2.29) \cdot 10^{16}$ | $0.00037 \pm 0.00085$ | $0.00047 \pm 0.00001$ | 12 |
| 2 | $(6.79 \pm 0.15) \cdot 10^{16}$ | $(9.25 \pm 4.70) \cdot 10^{16}$ | $0.00344 \pm 0.00175$ | $0.00252 \pm 0.00006$ | 10 |
| 3 | $(4.27 \pm 0.04) \cdot 10^{17}$ | $(4.08 \pm 0.41) \cdot 10^{17}$ | $0.01518 \pm 0.00151$ | $0.01587 \pm 0.00016$ | 5 |
| 4 | $(1.00 \pm 0.02) \cdot 10^{18}$ | $(1.10 \pm 0.08) \cdot 10^{18}$ | $0.04092 \pm 0.00290$ | $0.03717 \pm 0.00036$ | 1 |

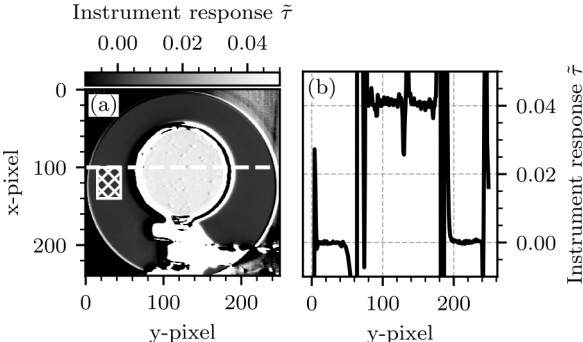

**Figure 9. (a)** The processed camera image for reference cell no. 4. The cell is in the center of the image. The circular structure behind it is the opening of the integrating sphere, in which a halogen lamp is placed as the light source of the experiment. The foreground shows the stand and clamp that are used to hold the cell in front of the integrating sphere. The in-cell region of the test cell shows a larger instrument signal than the background. The background region of our choice is marked with a patterned rectangle (left of the cell). **(b)** The instrument signal plotted along a vertical cross section through the middle of the test cell at $x = 100$ (see the dashed line in (a)). The region in the middle shows the enhanced signal within the cell. The strong peaks separating the background region and the in-cell region are generated by the frame of the cell. The strong structure that can be seen in the middle of the cell is due to condensation on the inside of the cell or similar imperfections of the experimental setup.

applied using the reference images according to eq. (12). The camera measures a signal of $\tilde{\tau} = (4.09 \pm 0.29) \cdot 10^{-2}$ in the in-cell region of the test cell. Using the calibration factor $k^{-1}$, a column density of $S = (1.10 \pm 0.08) \cdot 10^{18}$ molec cm$^{-2}$ was obtained. Within the uncertainty of the measurement this result coincides with that of the DOAS instrument, which measured a column density of $S = (1.00 \pm 0.02) \cdot 10^{18}$ molec cm$^{-2}$. Table 1 lists the column densities measured for each cell by the DOAS setup and the $NO_2$ camera. The measurements taken with the $NO_2$ camera show significant uncertainties. For cell no. 1, the relative uncertainty is as large as 231 % and the detection limits, ranging from $2.29 \cdot 10^{16}$ molec cm$^{-2}$ to $8 \cdot 10^{16}$ molec cm$^{-2}$, are larger than the prediction of the instrument model, which was $2 \cdot 10^{16}$ molec cm$^{-2}$ at 2 seconds of exposure. The reason for this deviation is the use of a different light source: While the instrument model assumed scattered sunlight as the light source,





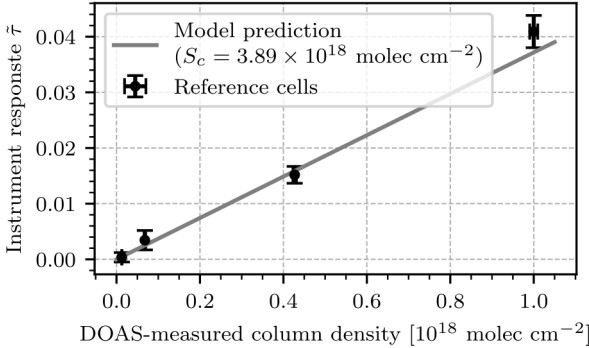

**Figure 10.** Scatter plot of the instrument response $\tilde{\tau}$ against the DOAS-measured column density of each test cell. The grey line shows the prediction of the instrument model with cell column density $S_c = 3.89 \cdot 10^{18}$ molec cm$^{-2}$.

a halogen lamp inside an integrating sphere was used for this experiment. The detection limit is mainly determined by the overall intensity of the light source, which is much lower for such a halogen lamp in the blue spectral range. This increased the statistical uncertainty of the measurement. Additionally systematic false signals were observed, which were not considered in the instrument model: Due to the small diameter of the test cells and the limited interior space of typical optical laboratories there are inevitable perspective shifts between the images of the two cameras, when they are oriented so that the test cells are in the center of their FOVs. Small dust particles on the test cell or condensed droplets on its inside can then introduce false signals. In order to smooth out these false signals, the images were convoluted with a rectangle filter of the same size as the average diameter of the observed structures. Table 1 lists the chosen filter size for each cell. The filter sizes were chosen differently for each cell, because larger cells require less smoothing. The cell image shown in Fig. 9 required no smoothing at all (which corresponds to a filter size of 1 pixel). Figure 10 shows a scatter plot of the instrument response of the $NO_2$ camera against the column density measured with the DOAS setup for each test cell. Additionally, the prediction of the instrument model (see sect. 2.2) with cell column density $S_c = 3.89 \cdot 10^{18}$ molec cm$^{-2}$ is plotted. The resulting instrument responses to the test cells are in very good agreement with the instrument model, with an average relative deviation of 18.2%. Model and measurement coincide for all test cells within the uncertainties of the measurement. Given the overall good agreement between the DOAS instrument, the $NO_2$ camera and the instrument model, we take these results as proof-of-concept.

## 4.2 Measuring the emissions of the coal power plant Großkraftwerk Mannheim

### 4.2.1 Setup and methodology

We report measurements taken at the Großkraftwerk Mannheim (GKM) with the $NO_2$ camera and a MAX-DOAS instrument. The GKM is a power plant located in Mannheim, Germany, which generates electricity based on burning of bituminous coal. It is one of the largest power suppliers of south-west Germany. The European Pollutant Release and Transfer Register (E-PRTR) lists an emission of 2,890,000 kg of $NO_x$ in 2017 (see The European Commision (2017)). The $NO_2$ camera was set up at





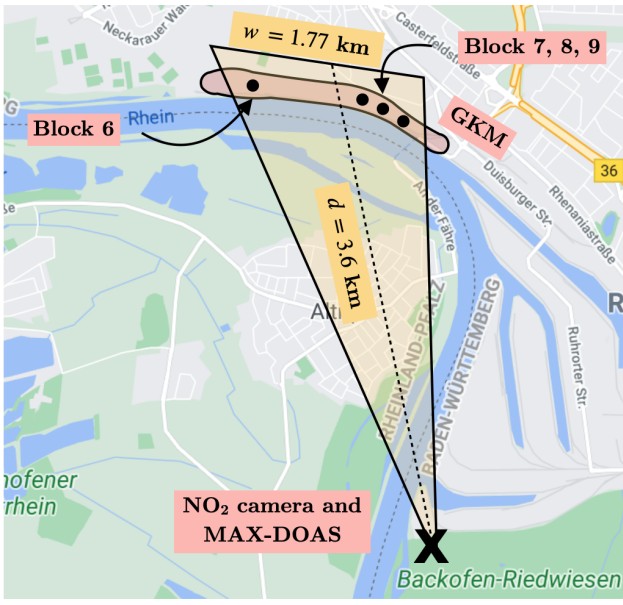

**Figure 11.** The GKM measurement in birds-eye perspective (from Google Maps, © Google Maps 2021). The instruments were set up at Backofen-Riedwiesen, 3.6 km south of the GKM and positioned, so that the emission of block 7 was in the middle of the FOV.

Backofen-Riedwiesen, 3.6 km south of the GKM (at 49.417745° N, 8.505917° W) on 26 April 2021. The sky was cloud-free on that day. The FOV of the camera at 3.6 km distance is approximately 1.77 km wide and 1.10 km high. However, it was
decided to decrease the read-out time of the camera modules by using windowing (see sect. 3). Therefore the true FOV was reduced to 1.22 km width and 0.53 km height. The camera was positioned, so that the plume emitted by GKM block 7 was in the center of the FOV. The optical axes of the two cameras were aligned, so that no shifts between their images were visible. The MAX-DOAS was set up to perform continuous elevation scans west (left) of the chimney of block 7. The $NO_2$ camera started recording images at 08:44 UTC+2. The MAX-DOAS instrument started scanning at 10:15 with a delay due to technical
issues. At regular intervals reference images of the sky at $45\,^\circ$ elevation angle were recorded.

During the measurement the camera with the empty cell recorded with an exposure time of $t_{\exp} = 2.688$ ms and the camera with the $NO_2$ cell recorded with an exposure time of $t_{\exp,c} = 11.027$ ms. Additionally, the cameras had a read-out time of 10 ms. The exposure times were chosen, so that the camera sensors were read out, once they were saturated to about 50 %. In order to increase image rate and reduce data volume, 100 consecutive frames were averaged, and these averages were saved.
We refer to them as images consisting of 100 frames. This way an image acquisition time of 2 seconds per 100 frames was achieved. The reference images were recorded in the same manner, although with exposure times $t_{\exp,\mathrm{ref}} = 5.765$ ms and $t_{\exp,c,\mathrm{ref}} = 22.895$ ms. This procedure yielded a total of four images $J$, $J_c$, $J_{\mathrm{ref}}$, and $J_{c,\mathrm{ref}}$. The resulting instrument signal image was then computed according to eq. (12), where all arithmetic operations and the logarithm were applied pixel-wise. In order to obtain sensible results, a few corrections had to be applied:





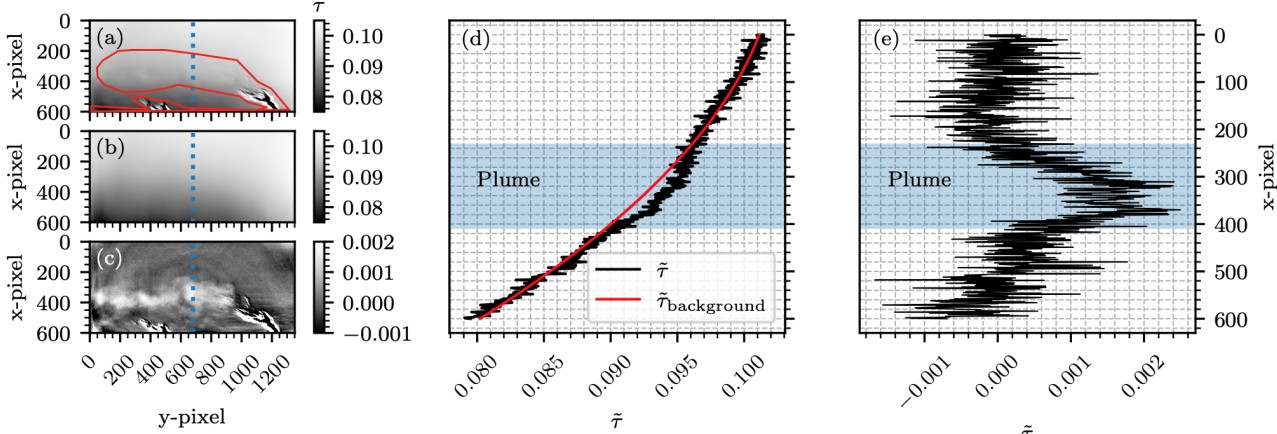

**Figure 12. (a)** Camera image of the GKM measurement (26 April 2021) without subtraction of the background fit. The plume signal is faintly visible between $x = 200$ and $x = 400$. The red solid line shows the outline of our manual definition of the in-plume region. **(b)** The background fit to the resulting off-plume region, extrapolated to the entire image. A polynomial of degree $n = 2$ was used as the fit function. **(c)** The instrument signal image obtained upon subtraction of the background fit. The plume signal is now clearly visible. **(d)** A plot along the vertical plume cross sections of image (a) and (b), indicated by the blue dotted vertical lines at $y = 660$. The solid line shows the original instrument signal $\tilde{\tau}$ along that vertical line without subtraction of the background fit. The red line shows the background signal obtained via the fit routine along that vertical line. **(e)** A plot along the vertical plume cross section at y = 660 of image (c), which demonstrates that the plume signal becomes visible in the residual upon subtraction of the background fit.

Firstly, the logarithm of the exposure time ratio

$$r = \ln\left(t_{\exp,c} \cdot t_{\exp,\mathrm{ref}}\right) - \ln\left(t_{\exp,c,\mathrm{ref}} \cdot t_{\exp}\right) \tag{28}$$

was subtracted in order to account for the fact that all four images were acquired with different exposure times.

      Secondly, a background image $\tilde{\tau}_{\mathrm{background}}$ was subtracted, for which the procedure and reasoning is described in the following. The background image was obtained by fitting a 1D-polynomial of degree $n$ to each column of a manually selected
set of background pixels, obtained by using a free-hand selection tool on the images. This was required, because the camera signal images showed large signal gradients across the FOV. We suspect that these gradients are a side-effect of the flat field correction, possibly because the sky, against which the reference images were taken, is generally not radiometrically uniform. An exemplary background correction procedure with $n = 2$ is shown in Fig. 12. The original signal image without background correction, as well as our manual choice of the plume and off-plume regions are displayed in subfigure (a). Subfigure (b)
shows the background fit on the basis of that choice. Subfigure (c) shows the resulting instrument signal image, with a clearly visible plume signal. Panel (d) shows, that for an exemplary column at $y = 660$, the background fit tailored very closely to the off-plume region and left a residual in the plume region. Subtraction of the fit made the plume signal visible in the residual,



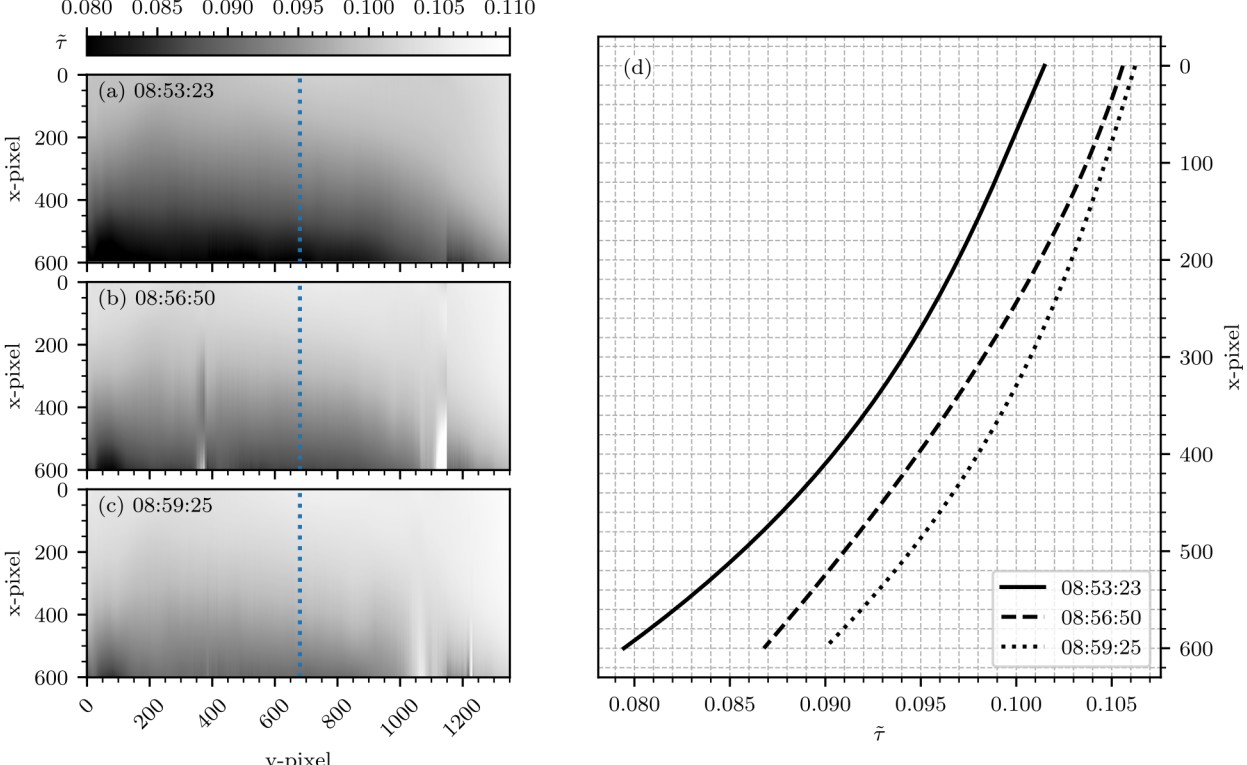

**Figure 13.** Temporal variance of the background images obtained from the background fit routine described in sect. 4.2.1. **(a)**, **(b)**, and **(c)** show the background fit to three images acquired at 08:53:23, 08:56:50, and 08:59:25 respectively. Panel **(d)** shows plots of the background signal along the blue dotted vertical lines at $y = 660$ in (a), (b), and (c). Together, the figures demonstrate the temporal variability in both magnitude and shape of the background signal.

which can be seen in panel (e). A weak temporal dependence of the background image was observed, possibly due to changes in the relative position of the sun (see Fig. 13).

Thirdly, a scalar signal offset $\tilde{\tau}_0$ was subtracted. The purpose of this correction was to account for slight variations of $S_c$ over the course of the measurement. The instrument model in sect. 2.2 showed, that the instrument signal $\tilde{\tau}$ is sensitive to the cell column density $S_c$, which again is expected to vary with ambient temperature and irradiance. For studying time-series it is important that this effect is accounted for, i.e. the signal in an off-plume reference region is forced to remain constant, which can be achieved by subtraction of a suitable offset $\tilde{\tau}_0$. Here, $\tilde{\tau}_0$ was computed by averaging the signal $\tilde{\tau}$ over a small rectangle

in the off-plume region (the patterned rectangle in Fig. 14) for each image individually.

    Finally the resulting signal images were multiplied with the calibration factor $k^{-1}$, which was obtained from the instrument model (see sect. 2.2). This required knowledge of $S_c$. $S_c$ was therefore estimated according to eq. (25), considering the same background rectangle as in the calculation of $\tilde{\tau}_0$ and a value of $S_c = (2.72 \pm 0.04) \cdot 10^{18}$ molec cm$^{-2}$ was obtained. With all

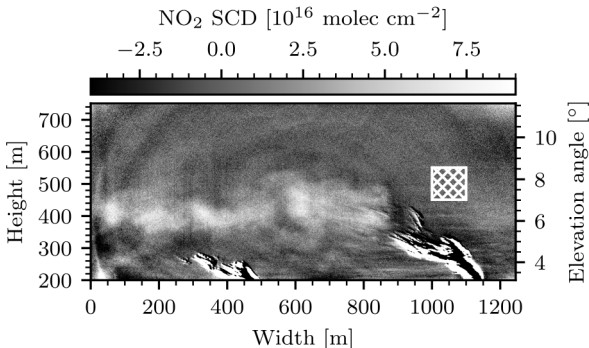

**Figure 14.** The first image of the measurement. For this image 6 individual images were averaged, which amounts to 12 seconds of total exposure. The center of the image shows the positive $NO_2$ plume signal of approximately $5 \cdot 10^{16}$ molec cm$^{-2}$. The patterned rectangle marks our choice for the off-plume region, used to calculate the column density in the gas cell of the instrument $S_c$, the signal offset $\tilde{\tau}_0$, and the detection limit $\Delta S$. At the point of emission (i.e. at widths of around 400 and 1000 m) the plume was in a fully condensed phase, which, due to optical misalignment of the cameras of the instrument towards the corners of the FOV, generates strong false signals.

corrections included, a single camera image was computed via

$$S = k^{-1}(S_c) \cdot \left( \ln\left( \frac{J_c \cdot J_{\text{ref}}}{J \cdot J_{c,\text{ref}}} \right) - r - \tilde{\tau}_{\text{background}} - \tilde{\tau}_0 \right) \tag{29}$$

where each pixel value $S_{(i,j)}$ corresponds to the $NO_2$ slant column density (SCD) measured at pixel $(i,j)$.

### 4.2.2 Evaluation of an individual camera image

Figure 14 shows the first camera image of the series, calculated according to eq. (29). To obtain this image, the first 6 consecutive images of the series were averaged. A background fitting routine with polynomial degree $n = 2$ and the same fit mask as displayed in Fig. 12 (a) were used (the choice of this fit mask is discussed further at the end of this section). A positive $NO_2$ plume signal equalling approximately $5 \cdot 10^{16}$ molec cm$^{-2}$ was observed to be emitted from the chimney of block 7. At the point of emission, i.e. directly above the chimney (at width = 1000 m), the plume was in a fully condensed phase and the instrument signal image shows structures of strong negative and positive signal. This effect can be explained as a consequence of the optical setup inside the instrument: The optical axes of the two cameras inside of the instrument were adjusted, so that there was no displacement of the imaged objects (i.e. the uncondensed part of the plume) in the center of the FOV. However displacements towards the corners of the FOV could not be avoided. These displacements manifest as strong false signals, when the signal ratio of the two cameras is computed. Given that in this measurement they occurred in an image region of low interest, they were deemed as unavoidable and not concerned with any further.

To obtain the $NO_2$ SCDs and the diameter $d$ of the plume systematically, each column of the $NO_2$ camera signal image was considered as an individual vertical cross section through the plume. It was observed that the shapes of the measured $NO_2$ SCDs along these cross sections coarsely followed that of a Gaussian. Figure 15 (a) shows this observation for an



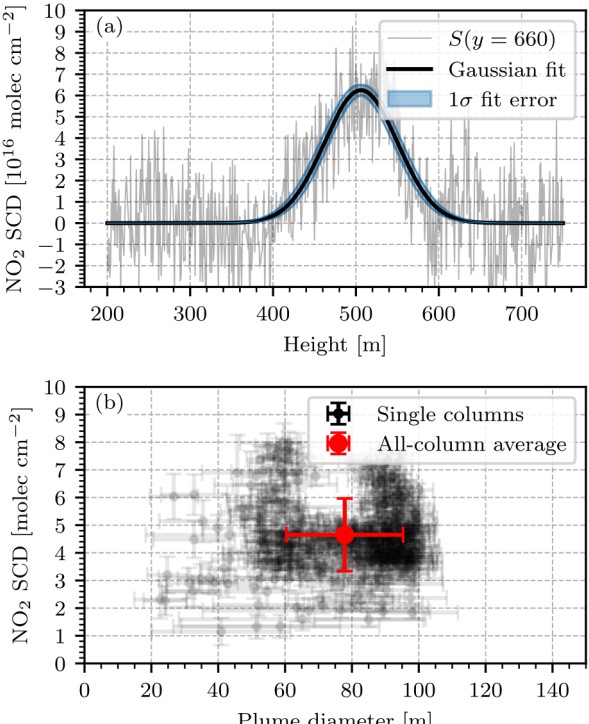

**Figure 15.** Evaluation of the camera image shown in Fig. 14. **(a)**: Plot of the measured $NO_2$ column density along the vertical plume cross section along $y = 660$ with a Gaussian fit. **(b)**: Scatter plot of the plume $NO_2$ column densities and diameters obtained from the camera image shown in Fig. 14 by fitting a Gaussian to each column of the image. The transparent black scatter points represent the single columns of the image, in which the fit quality criteria described in sect. 4.2.2 were met. The red scatter point in the center represents the average over all columns.

exemplary column at y-pixel 660. To each image column $i$, a Gaussian with amplitude $A_i$, mean $\mu_i$ and standard deviation $\sigma_i$ was fitted. The $NO_2$ slant column density and the diameter of the plume at column $i$ were then associated with $A_i$ and $2 \cdot \sigma_i$ respectively. Columns for which the fit routine did not converge well were ignored. This was considered the case when either

the fit failed to converge entirely or the retrieved fit parameters were outside a realistic range ($A_i = S_i > 8 \cdot 10^{16}$ molec cm$^{-2}$ or $2 \cdot \sigma_i = d_i > 100$ m), which was the case for approximately 50 % of the columns. The resulting $NO_2$ SCDs and plume diameters are shown in Fig. 15 (b). The ensemble of all column fits allows to calculate an average in-plume $NO_2$ SCD of $S = (4.74 \pm 1.21) \cdot 10^{16}$ molec cm$^{-2}$ and an average plume diameter of $d = (78 \pm 17)$ m. These values are represented by the red marker in Fig. 15 (b). It is necessary to discuss the uncertainties of such an evaluation procedure. It was explained in

sect. 2.2 (see specifically eq. (16)) that the measurement has an intrinsic uncertainty $\Delta\tilde{\tau}$ of the uncalibrated camera signal due to the Poissonian error of photon counting. This uncertainty propagates directly onto the $NO_2$ SCDs, that are obtained upon calibration of the instrument using $k^{-1}(S_c)$ as described in eq. (29) and was estimated by computing the standard deviation





**Figure 16.** Comparison of the results from different variants of the background fitting procedure as described in sect. 4.2.1. The left two columns **(a-h)** show the procedure for the freehand mask used in sect. 4.2.1, but with different polynomial degrees of up to $n = 4$. The right two columns show the same procedure for a fit mask that covers the entire FOV of the camera images. The results are summarized in Table 2.





**Table 2.** Summary of results from different variants of the background fitting procedure as described in sect. 4.2.1 and shown in Fig. 16. The full FOV fit mask yields smaller plume SCDs and diameters than the freehand mask. $n = 2, n = 3$, and $n = 4$ yield similar results for both fit masks.

| Subfigure | $n$ | Fit mask | Average plume SCD [$10^{16}$ molec cm$^{-2}$] | Average plume diameter [m] | Successful fits |
|-----------|-----|----------|-----------------------------------------------|----------------------------|-----------------|
| (c-d) | 2 | Freehand | $4.74 \pm 1.22$ | $78 \pm 17$ | 439/900 |
| (e-f) | 3 | Freehand | $4.70 \pm 1.01$ | $82 \pm 13$ | 502/900 |
| (g-h) | 4 | Freehand | $5.03 \pm 1.01$ | $77 \pm 18$ | 480/900 |
| (k-l) | 2 | Full FOV | $3.54 \pm 1.02$ | $47 \pm 11$ | 538/900 |
| (m-n) | 3 | Full FOV | $3.26 \pm 1.17$ | $38 \pm 11$ | 440/900 |
| (o-p) | 4 | Full FOV | $2.97 \pm 0.92$ | $36 \pm 9$ | 477/900 |

of the measured $NO_2$ SCDs in an off-plume region of a camera image, e.g. the patterned rectangle in Fig. 14. A value of $\Delta S = 1.89 \cdot 10^{16}$ molec cm$^{-2}$ was obtained. This is the detection limit of the instrument prototype. In the next step the plume

SCDs and diameters were obtained in a Gaussian fit routine for the vertical plume cross sections of all image columns. For a single column $i$, this introduced additional uncertainties $\Delta A_i$, $\Delta \mu_i$, and $\Delta \sigma_i$, which were given by the covariances of the fit parameters of that column. These uncertainties propagate into those of the means over all columns, producing the uncertainties used above ($\Delta S = 1.21 \cdot 10^{16}$ molec cm$^{-2}$ and $\Delta d = 17$ m). Finally the uncertainties of the background fitting routine as described in sect. 4.2.1 were investigated. The camera image shown in Fig. 14 was calculated according to eq. (29), where

$\tilde{\tau}_{\text{background}}$ was computed using a polynomial of degree $n = 2$ and the same fit mask as displayed in Fig. 12 (a). Given that this choice of $n$ and the fit mask are subject to our personal assessment, it was investigated, how much the obtained $NO_2$ SCD and diameter of the plume vary with different choices of $n$ and the fit mask. Figure 16 shows the results of this analysis. Subfigures (a-h) show the process of the background fitting routine using the freehand fit mask that was described earlier. Subfigure (c, e, g) show the resulting background images $\tilde{\tau}_{\text{background}}$ for $n = 2, 3, 4$ respectively. Subfigures (d, f, h) show

the corresponding scatter plots of $NO_2$ SCD and plume diameters as obtained from the Gaussian fit routine. Subfigures (i-p) show the same procedure with a different fit mask, namely one that makes no assumptions of the plume position and covers the entire FOV. The case $n = 1$ was dismissed, seeing that the background signal is clearly not linear (see Fig. 12 and Fig. 13). Intercomparison of subfigures (d), (f), and (h) as well as (l), (n), and (p) shows, that for a given fit mask the average $NO_2$ SCD and plume diameter do not vary significantly with the choice of $n$. Using a full-FOV fit mask yields significantly

smaller average values of $NO_2$ SCD and plume diameter. Furthermore, image objects such as the condensed plumes at y-pixel 400 and 1200 lead to vertical fragments in the background image (see subfigure (j)). Overall, the background fitting procedure with $n = 2$ and a freehand selection of the plume as displayed in subfigure (c) and (d) seems to be a sensible choice, because the resulting background image does not suffer from vertical fragments and shows less signal variations in the off-plume region. In addition the fit is fastest to compute for $n = 2$. Table 2 contains a quantitative summary of these





findings and allows to estimate the uncertainty of the background fitting routine. The uncertainty of the $NO_2$ SCDs spans from $(2.97 - 0.92) \cdot 10^{16}$ molec cm$^{-2}$ = $2.05 \cdot 10^{16}$ molec cm$^{-2}$ to $(5.03 + 1.01) \cdot 10^{16}$ molec cm$^{-2}$ = $6.04 \cdot 10^{16}$ molec cm$^{-2}$. The mean is $4.04 \cdot 10^{16}$ molec cm$^{-2}$. Therefore, the overall uncertainty can be estimated as $\Delta S = 2 \cdot 10^{16}$ molec cm$^{-2}$. In analogy an uncertainty of $\Delta d = 34$ m for the plume diameter is obtained, which will be used throughout the rest of this chapter. With this method an estimate of the overall uncertainty of the evaluation is obtained, by including not only the statistical uncertainty 430 of the measurement (noisy data), but also the systematic uncertainty that is immanent to the evaluation method.

A series of camera images was assembled into a video (see video supplement), which shows the movement of the plume in wind direction from 08:53 to 09:05.

### 4.2.3 Optical flow and mass flux analysis

A mass flux analysis was carried out on the basis of image sequences. Given a camera image as shown in Fig. 14, the mass flux 435 through a vertical cross section of the plume can be computed as

$$F_m = \frac{M_{NO_2}}{N_A} \cdot v \cdot \int S(h) \, \mathrm{d}h \tag{30}$$

where $M_{NO_2} = 46.0055$ g mol$^{-1}$ is the molar weight of $NO_2$, $N_A = 6.022 \cdot 10^{23}$ mol$^{-1}$ the Avogadro number, $v$ the wind speed in horizontal direction and $S$ the column density, which is integrated along the vertical (height) axis. $v$ was obtained by running a Farnebäck optical flow retrieval (Farnebäck (2003)) on the in-plume region of consecutive camera images. The 440 optical flow was then divided by the time difference $\Delta t$ between the images. Figure 17 (a) shows the wind speeds associated with the camera image in Fig. 14. For this image and its successor, a mean horizontal wind velocity of $v = (1.48 \pm 0.39)$ m s$^{-1}$ was obtained. The average was considered over the plume region only, because in the still background the Farnebäck algorithm can not detect any flow and returns a wind speed of 0. Similar to the column-wise evaluation of the $NO_2$ SCD and plume diameter in sect. 4.2.2, the $NO_2$ mass flux was computed through each column separately, according to eq. (30).

Figure 17 (b) shows the $NO_2$ mass flux, obtained through the individual columns of the image that was displayed in Fig. 14, plotted against the distance travelled downwind from the point at which the fully condensed part of the plume ended (see Fig. 14 or Fig. 17 (a) at width = 840 m). This procedure yielded a mean mass flux of $F_m = (13.63 \pm 7.89)$ kg h$^{-1}$. The evaluation was extended to obtain average wind speeds and mass fluxes as a function of time. The results are displayed in Fig. 18. Subfigure (a) shows the mean horizontal wind speed and subfigure (b) shows the mean $NO_2$ mass flux. Over the observed 450 time frame from 08:55 to 9:25 UTC+2, an overall mean horizontal wind speed of $v = (0.94 \pm 0.33)$ m s$^{-1}$ and a $NO_2$ mass flux of $F_m = (7.41 \pm 4.23)$ kg h$^{-1}$ = $(64.5 \pm 36.8)$ tons yr$^{-1}$ were obtained.

A combination of several publicly available sources can be used to estimate a reference value for the $NO_x$ mass flux of the GKM, which can be compared to the value measured here. Of course, the $NO_2$ camera data only allow to compute the $NO_2$ mass flux, not the $NO_x$ mass flux. However, the large FOV of the camera covers a total distance of up to 1000 m downwind 455 from the point of emission. It can therefore be expected that the main chemical conversion processes (see eq. (R1), (R2), and (R3)) have reached equilibrium and the Leighton relationship is reached. In that case the mass fluxes of $NO_2$ and $NO_x$ should be of comparable magnitude.





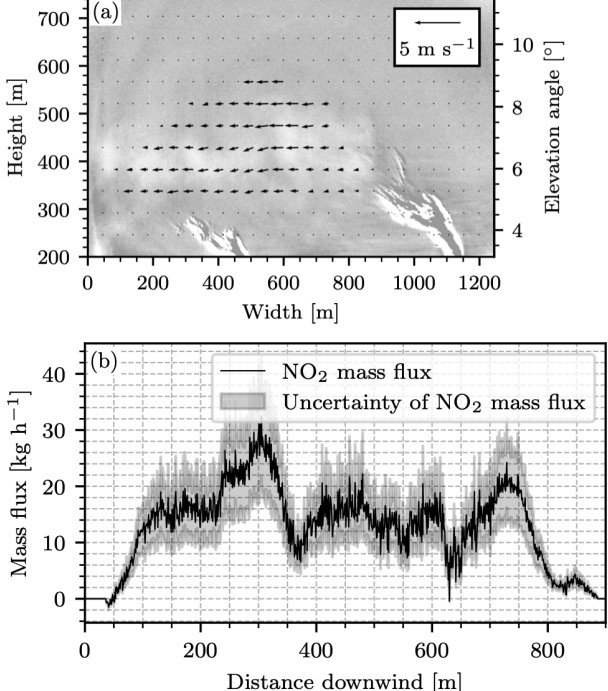

**Figure 17. (a)** Wind speeds determined from the camera image in Fig. 14 and its successor by application of the Farnebäck algorithm. The wind field is displayed as a vector field in the plume region. **(b)** $NO_2$ mass flux obtained from the camera image shown in Fig. 14 and the wind field shown in (a). The mass flux was plotted against the distance downwind, measured from the point, where the fully condensed part of the plume ends (at a width of 840 m in (a)).

The Fraunhofer Institute for Solar Energy Systems (ISE) reports, that the GKM was producing 70.6 MW at 09:00 on the day of measurement (see Fraunhofer Institute for Solar Energy Systems (2021)). The European Pollutant Release and Transfer
Register lists an $NO_x$ emission of the GKM of 2890 tons yr$^{-1}$ in 2017 (The European Commision (2017)). The business report of the GKM of the same year states a mean power production of 1119 MW (Großkraftwerk Mannheim Aktiengesellschaft (2018)). Therefore the GKM should have been running at approximately 6.3 % of its average power. Assuming that the $NO_x$ emission scales linearly with the power produced, a $NO_x$ mass flux of $F_m = 182$ tons yr$^{-1}$ is expected. The mean mass flux obtained from the camera data is significantly lower, and amounts only to about one third of this reference value. Given that
the reference is a $NO_x$ mass flux and the $NO_2$ camera can only detect the $NO_2$ mass flux, such deviations are expected.

It should be taken into account that this analysis contains two further uncertainties: Firstly, although the most recent available data were used, there may be differences in the reference values between 2017 and 2021 (e.g. total mass of yearly emitted $NO_x$ or mean power production). The E-PRTR data show a decline in total yearly emitted $NO_x$ from 2007 to 2017 and it can be expected that this trend has continued until 2021. It should be taken into account, that a comparison between a mean
flux observed in a time frame of 30 minutes and a yearly average reference flux is hardly indicative for the accuracy of our





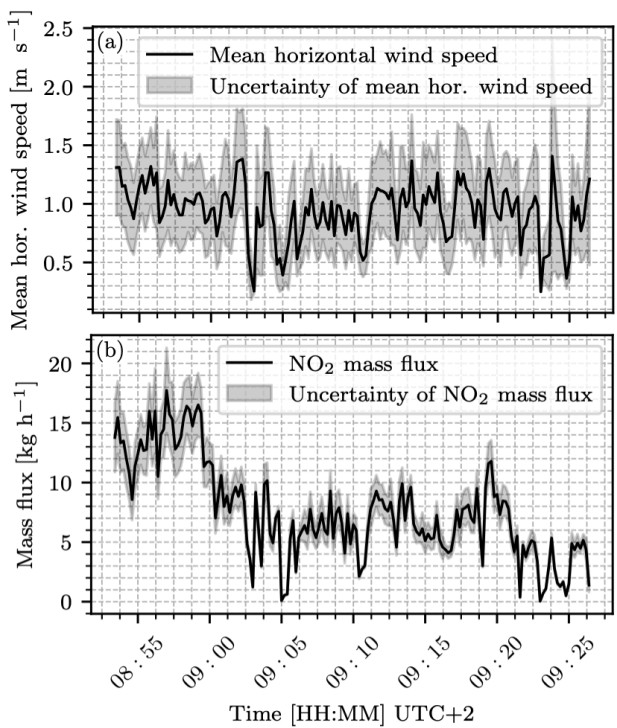

**Figure 18.** Evaluation of average horizontal wind speeds and $NO_2$ mass fluxes based on the camera images recorded on 26 April 2021 between 08:55 and 9:25. **(a)** The mean horizontal wind speed, as obtained from the Farnebäck algorithm on consequtive image pairs. **(b)** The resulting mean $NO_2$ mass fluxes, calculated according to eq. (30).

measurement. Secondly, GKM block 7, of which the emitted plume column densities were used for this analysis, was not the only active block at the time of the measurement. During the measurement, emissions from GKM blocks 6 and 8 were observed as well, but the FOV of the $NO_2$ camera was too small to record the plumes emitted from all blocks simultaneously. It is plausible to assume additional emissions of $NO_2$ from GKM block 6 and 8, which could not be examined on the basis of

our measurement.

Although the discussed uncertainties do not allow for a definite conclusion on the overall accuracy of the mass flux analysis, we present the results as a demonstration that flux analyses on the basis of image data with high spatio-temporal resolution are a feasible concept.

### 4.2.4 Estimation of $[NO_2]/[NO_x]$ ratios

The camera images can be used to investigate the conversion of NO to $NO_2$ by the reaction of NO with ambient ozone (see eq. (R1)) and direct oxidization by molecular oxygen (see eq. (R2)). The $NO_2/NO_x$ ratio can be modelled according to Janssen



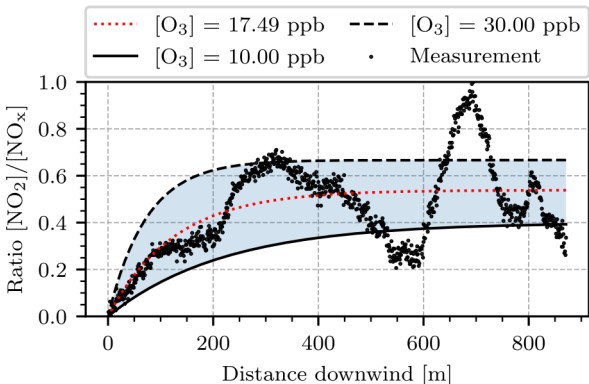

**Figure 19.** Plot of the $[NO_2]/[NO_x]$ ratio as a function of distance travelled downwind, measured from the point, where the fully condensed part of the plume ended (at a width of 840 m in Fig. 14). The black scatter markers represent the concentration ratio obtained on the basis of the camera data. The black solid and dashed lines show predictions of the Janssen model for different ozone mixing ratios and a wind speed of $v = 0.94 \, \text{m s}^{-1}$. The dotted red line is a fit of the Janssen model to the measured data points.

et al. (1988) by the formula

$$\frac{[NO_2]}{[NO_x]} = \left(1 - e^{-ax}\right) \cdot \left(\frac{[O_3]}{A + [O_3]}\right) \tag{31}$$

where $x$ is the distance downwind from the point of emission, and $[NO_2]$, $[NO_x]$, and $[O_3]$ denote the concentrations of $NO_2$,

$NO_x$, and $O_3$. The model has a parameter $a = k[O_3]/v$, where $k$ is the rate constant for the $NO + O_3 \rightarrow NO_2 + O_2$ reaction, and $v$ the wind speed, as well as another parameter $A = J/k$, where $J$ is the photodissociation frequency of $NO_2$. The rate constant $k(T)$ is temperature dependent. Lippmann et al. (1980) find the empirical relationship

$$k(T) = 4.3 \cdot 10^{-12} \cdot e^{-1598 \, \text{K}/T} \, \text{cm}^3 \, \text{molec}^{-1} \, \text{s}^{-1} \tag{32}$$

with temperature $T$. The photolysis frequency $J$ is often cited as approximately $J = 8 \cdot 10^{-3} \, \text{s}^{-1}$ in full sunshine (see e.g.

Platt and Kuhn (2019)), but varies strongly with irradiance (Parrish et al. (1983)). Figure 19 shows an approach to compare the camera measurements with the Janssen model. The parameters of the Janssen model are determined by the wind speed $v$, the ozone concentration $[O_3]$, the photodissociation frequency $J$, and temperature $T$. For the wind speed $v = 0.94 \, \text{m s}^{-1}$ was assumed, as obtained from the optical flow procedure in sect. 4.2.3. The remaining parameters (ozone concentration, photolysis frequency, and temperature) were obtained by fitting the Janssen model to the measured data points. For this the

first 1000 images of the series were averaged (this amounts to a time window from 08:45 to 09:30). Then the vertical integrals of the plume SCD $\int S(h) \, dh$ were computed for each individual column, like in the mass flux analysis in sect. 4.2.3. The concentration ratio $[NO_2]/[NO_x]$ associated with each image column was obtained by normalizing this set of integrated SCDs into the interval $[0, 1]$. This is in accordance with the Janssen model, which predicts an initial concentration ratio of 0 with an exponential convergence towards a concentration ratio of $\leq 1$, depending on the model parameters. Figure 19 shows these



obtained ratios as black dots, plotted against the distance downwind, measured from the point, where the fully condensed part of the plume ended (see Fig. 14 at width = 840 m). By running a least-squares fit routine, an ozone mixing ratio of $[O_3] = 17.49$ ppb, a temperature of 13.85 °C and a photolysis frequency of $J = 6.4 \cdot 10^{-3}$ s were obtained. As a reference, the closest ground-based air quality measuring station (Mannheim-Nord, DEBW005) measured an ozone mixing ratio of 26.79 ppb at 09:00 (Landesanstalt für Umwelt Baden-Württemberg, 2021). However, it should be taken into consideration, that such

ground-based measurements may not yield representative values for 200 - 500 m altitude. Moreover, temperatures of up to 17.3 °C were reported in Mannheim for the day of our measurement (Deutscher Wetterdienst (2021)). Parrish et al. (1983) report similar values of $J$ at solar zenith angles of approximately 60 °, while the solar zenith angle at the beginning of our measurement was 77.7 °.

Overall, the data points in Fig. 19 coarsely resemble the shape of the Janssen model. However, they oscillate around the

prediction of the best fit (red dotted line in Fig. 19). The cause of these oscillations is possibly the alignment of the optical axes of the cameras inside the instrument. It was explained earlier, that the camera axes were aligned so that no shifts occur in the center of the FOV due to the displacement of the two cameras. However, shifts towards the corners of the FOV are then inevitable. It was observed, that such shifts typically lead to patterns of consecutively increased and decreased false signal in the signal ratio image. The plateau after 400 m of downwind distance agrees with Janssen models assuming ozone mixing

ratios of 15 - 30 ppb. Although such mixing ratios are relatively low for typical polluted urban areas, they are within a realistic order of magnitude. Overall, the obtained fit parameters agree well with the reference values we have listed. It should be taken into consideration that more recent studies have found initial $NO_2/NO_x$ concentration ratios of 5 - 10 % to be more realistic for the emission from most combustion processes (see e.g. Kenty et al. (2007); Carslaw (2005)). This is neglected by the Janssen model, which predicts an initial $NO_2/NO_x$ ratio of zero. Furthermore, as discussed in sect. 4.2.2, the $NO_2$ camera is incapable

of measuring the $NO_2$ SCD of the plume directly after its emission, when it is still in a fully condensed phase (see Fig. 14). Figure 19 shows the concentration ratio against the distance downwind, which is measured from the point, where the fully condensed part of the plume ended (at a width of 840 m in Fig. 14). The evaluation shown here neglects the plume chemistry of this early phase. To conclude, a crucial uncertainty is the mapping of the column-wise vertical SCD integrals onto the interval $[0, 1]$ on both ends: At the lower end, near the point of emission, the concentration ratio is unmeasurable, due to the phase

of the plume. At the upper end, far downwind, the mapping could be slightly off due to the oscillations of the measured data points. However, we notice good agreement between the obtained fit parameters and the reasonably picked reference values listed earlier in this section.

### 4.2.5 Comparison of camera images and MAX-DOAS elevation scans

A comparison between the $NO_2$ column densities measured by the $NO_2$ camera and the results from elevation scans with the

MAX-DOAS instrument was carried out. The MAX-DOAS instrument and the $NO_2$ camera operated simultaneously from 10:15 until 10:50 on 26 April 2021.

Figure 20 (a) shows the results of an exemplary MAX-DOAS elevation scan, taken from 10:15 to 10:17. The differential $NO_2$ SCD at each elevation angle was obtained by fitting the absorption cross sections of $NO_2$, $O_4$, $O_3$, and a Ring spectrum



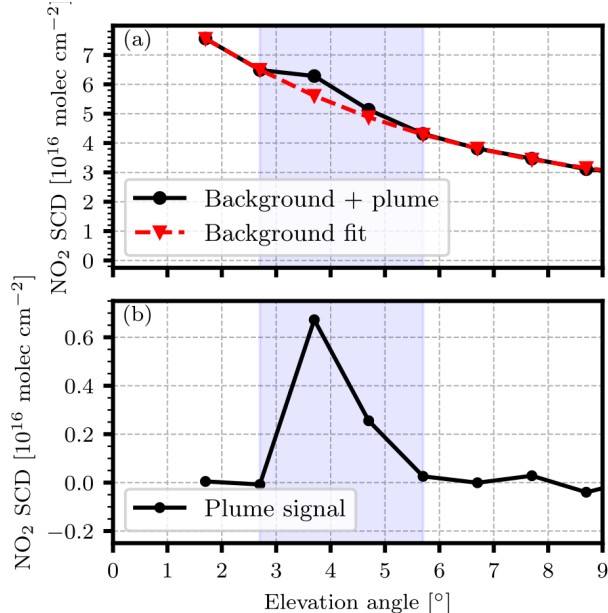

**Figure 20.** Separation of plume signal and background signal in an exemplary elevation scan with the MAX-DOAS instrument, recorded at 26 April 2021, from 10:15 to 10:17. **(a)** The $NO_2$ signal obtained from the DOAS analysis and a background fit to it. The elevation angles 2-4 ° were excluded from the fit to achieve consistency with the background estimation routine of the camera data (see sect. 4.2.1). **(b)** The resulting plume signal, which is obtained by subtracting the background fit (red line in (a)) from the total signal (black line in (a)). The blue-shaded region is intended to help compare subfigures (a) and (b).

to the measured optical depth in a fit window from 400.6 to 425 nm. Spectra recorded at 90 ° were used as reference. Elevation
scans taken over polluted regions typically show a strong anthropogenic $NO_2$ background, caused by the combined emission of
the many $NO_2$ sources (e.g. fuel burning in traffic, in households or on industrial sites) in these regions. For reference, similar
backgrounds were measured e.g. by Manago et al. (2018) and Peters et al. (2019). In order to extract the $NO_2$ signal attributed to
the emission of GKM block 7, the anthropogenic background signal (the red line in Fig. 20 (a)) was estimated and subtracted
from the total signal (the black line in Fig. 20 (a)). The procedure is similar to the background fit described in sect. 4.2.1,
although here the intention was to separate two sources of true $NO_2$ signal (plume and background), whereas the background
fit procedure for the camera data intended to additionally separate plume signal and false signal due to imperfections of the
optical setup. For ideal comparability the DOAS background signal was extracted using the same background estimation
routine that was used for the camera data (see sect. 4.2.1). Here a cubic function (polynomial of degree $n = 3$) was used for
both camera and DOAS data. Additionally, the elevation angles that were masked out in the camera data by choice of the fit
mask were masked out in the DOAS fit as well. In practice, this means that here the elevation angles 2-4 ° were excluded from
the fit. The plume signal was then extracted by subtraction of the obtained background signal from the total signal, as shown
in Fig. 20 (b), leaving a clear signal spike at an elevation angle of 3 °.



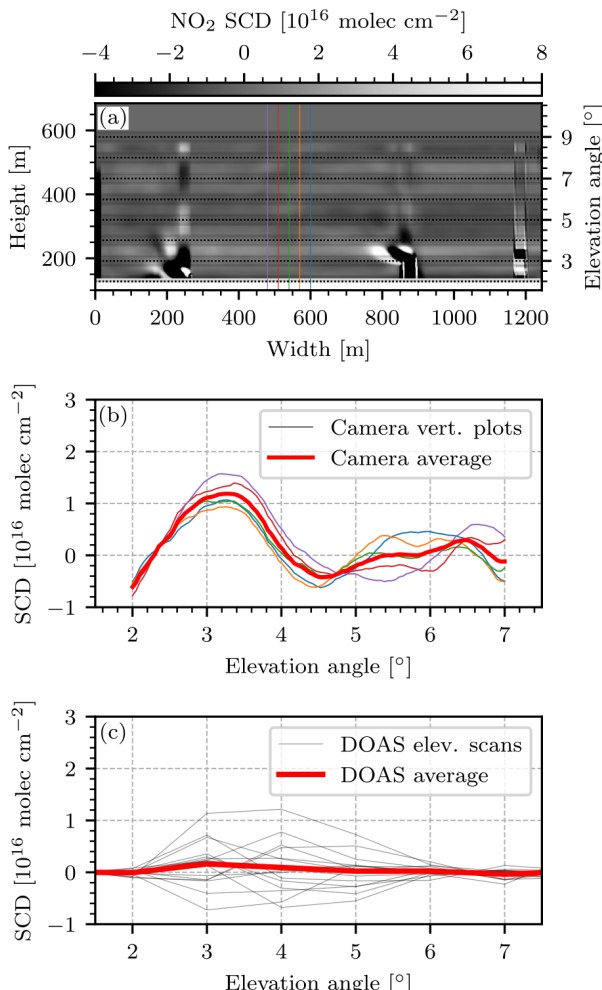

**Figure 21.** Comparison between the camera images and MAX-DOAS elevation scans on 26 April 2021 from 10:15 to 10:50. **(a)** The camera data, averaged over the specified time frame. The camera FOV is slightly different from that in Fig. 14, because the instrument was re-oriented in between the recording of reference images against the sky. In order to account for the FOV of the MAX-DOAS instrument of $\pm 0.2\,°$, the camera images were smoothed with a Gaussian filter of corresponding size (32 pixels). The coloured vertical lines mark the vertical plume cross sections, through which the camera data is sampled. The black horizontal lines mark the FOV of the MAX-DOAS instrument. **(b)** Plot of the camera data along the vertical plume cross sections marked in (a). **(c)** The elevation scans of the MAX-DOAS instrument. Each black line corresponds to a full elevation scan.

In order to compare camera images with MAX-DOAS elevation scans, the spatio-temporal resolution of the camera data was adjusted to that of the MAX-DOAS instrument. For this purpose the average over all camera images recorded in the time range
from 10:15 to 10:50 was computed and smoothed with a Gaussian filter with a standard deviation of $\sigma = 16$ pix or filter size of





32 pix, corresponding to the MAX-DOAS instrument's FOV of $\pm\,0.2\,°$. Although this drastically reduced the spatio-temporal resolution of the $NO_2$ camera data, it was nonetheless still higher than that of the MAX-DOAS instrument. This is because the MAX-DOAS instrument only samples one elevation angle at a time, while the $NO_2$ camera captures a full-frame image, yielding a higher overall information content. Also the MAX-DOAS measurements do not continuously cover the range of

elevation angles, because the FOV is smaller than the steps of the elevation scan. The resulting camera image is displayed in Fig. 21 (a) along with some coloured vertical and horizontal lines indicating the regions in which the comparison to the MAX-DOAS elevation scans was carried out (see figure description). Figure 21 (b) shows plots of the camera signal along the vertical plume cross sections indicated by the coloured vertical lines in Fig. 21 (a). Compared to the results shown in Fig. 14, the overall plume signal is much lower, peaking at around $1.5 \cdot 10^{16}\ \mathrm{molec\ cm^{-2}}$ (see purple line in Fig. 21 (b)). The

average plume signal amounts to around $1 \cdot 10^{16}\ \mathrm{molec\ cm^{-2}}$ (see thick red line in Fig. 21 (b)). The plume was located at a low elevation angle of approximately $3.3\,°$. Figure 21 (c) shows the results of the MAX-DOAS elevation scans. These were obtained by repeating the background fitting routine displayed in Fig. 20 for all elevation scans individually. The elevation scans show enhanced $NO_2$ signals with peaks of up to $1 \cdot 10^{16}\ \mathrm{molec\ cm^{-2}}$ in an elevation angle range of $3 - 6\,°$. However, for some DOAS elevation scans, a negative plume signal of up to $-0.8 \cdot 10^{16}\ \mathrm{molec\ cm^{-2}}$ is obtained after subtraction of the

background fit. As a consequence, in the average DOAS elevation scan (see thick red line in Fig. 21) the measured $NO_2$ SCD is largely reduced. These deviations from the camera measurements require discussion.

In order to compare the elevation angles, at which both instruments registered the target plume signal, an elevation calibration method is required for both instruments. The elevation calibration for the $NO_2$ camera is straightforward. Using the chimney height of 200 m and a distance between the $NO_2$ camera and the GKM of 3.6 km, the chimney top is at an elevation angle

of $\arctan(200/3600) \approx 3.2\,°$. Both chimney height and distance to the scene are known very accurately and thus we take this calibration as reliable. MAX-DOAS instruments are typically calibrated by scanning the horizon in small elevation steps and identifying the elevation angle of $0\,°$ by the sudden jump of measured intensity that is expected to occur at the horizon line. From the perspective of our measurement location, the rooftops of the GKM buildings were reaching over the horizon, meaning that the MAX-DOAS calibration has registered the intensity jump that occurred between the sky and the rooftops at

a height of 120 m or $1.9\,°$ elevation (as opposed to the horizon at 0 m height or $0\,°$ elevation). This reference allows for full elevation calibration of the MAX-DOAS instrument. Because time was running short, the horizon scan was performed with relatively coarse elevation steps of around $1\,°$. Combined with the extended FOV of $\pm 0.2\,°$ of the MAX-DOAS instrument, the uncertainty of the elevation angle can be assumed to be on the order of $1\,°$.

As explained, a difference in the overall magnitude of the measured plume signals is observed as well. While the $NO_2$

camera measures an average plume signal of approximately $1 \cdot 10^{16}\ \mathrm{molec\ cm^{-2}}$, the corresponding average values measured by the MAX-DOAS instrument are much lower. Two possible reasons can be identified for this deviation:

Firstly, the background fitting routine is a critical step for both instruments. It was shown in sect. 4.2.2, that the uncertainty of the overall plume signal is around $2 \cdot 10^{16}\ \mathrm{molec\ cm^{-2}}$. This uncertainty was mainly constituted by the detection limit of the instrument due to the uncertainty of photon counting, the variance in measured signal at different distances downwind and

the uncertainty of which background fit function to use (see Fig. 16). Here, because the camera data was averaged over a time




span of 35 minutes and smoothed with a Gaussian filter, the overall uncertainty was strongly reduced to approximately $\Delta S \approx 0.6 \cdot 10^{16}$ molec cm$^{-2}$ (see Figure 21 (b)), however the plume signal is also much smaller at approximately $1 \cdot 10^{16}$ molec cm$^{-2}$. With such large relative uncertainties, it can not be expected that the background fitting routine separates background signal and plume signal clearly. Instead it is much more likely, that a small portion of the plume signal is removed as well when

subtracting the background fit. The situation is quite similar for the MAX-DOAS instrument, where a plume signal on the scale of $1 \cdot 10^{16}$ molec cm$^{-2}$ must be separated from a background signal gradient on the scale of at least $5 \cdot 10^{16}$ molec cm$^{-2}$ (see Fig. 20). Figure 21 (c) demonstrates, that the distinction between plume and background signal is so uncertain, that a for a few elevation scans the routine even returns negative plume signal.

Secondly, and more importantly, the two instruments achieve spatio-temporal sampling of the plume on very different scales.

The NO$_2$ camera is capable of full-frame imaging with acquisition times on the scale of seconds and can measure the plume regardless of its position and shape, given a high enough plume column density. The MAX-DOAS on the other hand requires around 2 minutes for a single elevation scan of 14 individual elevation angles, meaning it effectively yields a point measurement every 8.5 seconds. The temporal sampling in the plume region is therefore rather low. Specifically, the MAX-DOAS samples the plume region for only around 20 % of the measuring time. It is very likely that during the elevation scans, the MAX-DOAS

FOV missed the plume center, where the highest SCD would be measured, or even missed the plume entirely. Additionally, variations of the vertical plume position during sampling may have further decreased the obtained signal or even led to negative signals.

Figure 21 (c) shows, that some MAX-DOAS elevation scans measured an enhanced signal at an elevation angle of 3.7 °, which is close to where the NO$_2$ camera located the plume at around 3.3 ° (see subfigure (b)). One of these MAX-DOAS scans,

scan no. 9, was taken from 10:20 to 10:22. Figure 22 shows a direct comparison between that elevation scan and the camera results, averaged over that respective time frame (as opposed to Fig. 21 (a) and (b), which show the camera signal averaged over the entire measurement from 10:15 to 10:50). The dotted blue lines show the upper and lower edge of the uncertainty of the MAX-DOAS elevation calibration, here assumed to be $\pm 1$ °. Within the uncertainty, a reasonable agreement with the camera data is possible in the region around 3.3 ° elevation. Overall the data shown in Fig. 22 yields the best agreement between

MAX-DOAS and NO$_2$ camera results that we could find.

To conclude, the agreement between camera and MAX-DOAS data is fair at best. For future measurements the MAX-DOAS elevation scans should be carried out in smaller elevation steps and longer acquisition times per step to accommodate for its poor spatio-temporal resolution. If possible, emission plumes with higher NO$_2$ concentrations and less turbulent atmospheric conditions should be chosen.

**5 Conclusion**

We present a prototype of a novel NO$_2$ imaging instrument based on Gas Correlation Spectroscopy: the GCS NO$_2$ camera. It operates by recording images with two cameras, each with a gas cell (cuvette) in front of it, where one is filled with air and the other filled with a high concentration of NO$_2$. The instrument acquires images at high spatio-temporal resolutions



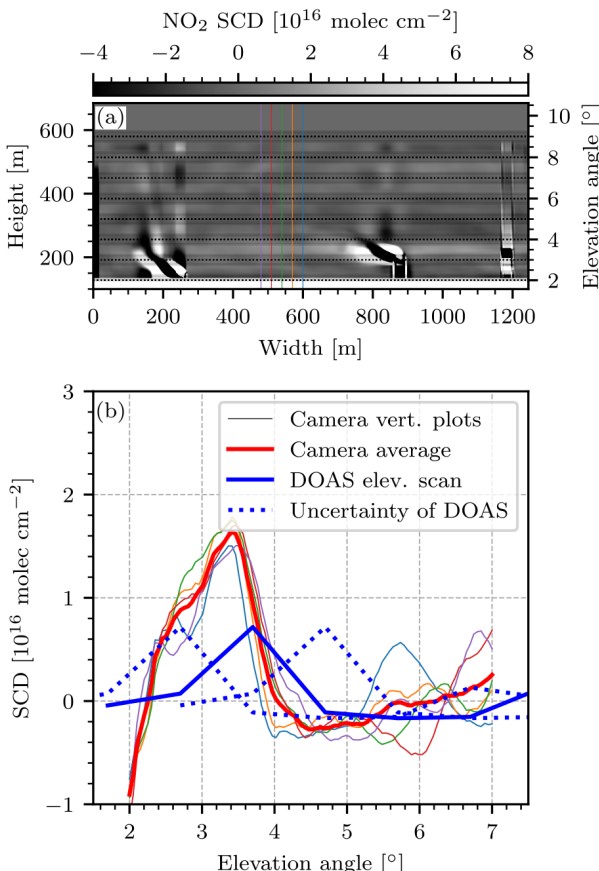

**Figure 22.** Comparison between the camera images and MAX-DOAS elevation scans on 26 April 2021 from 10:20 to 10:22. **(a)** The camera data, averaged over the specified time frame. The camera images were smoothed with a Gaussian filter of corresponding size (32 pixels). The coloured vertical lines mark the vertical plume cross sections, through which the camera data is sampled. The black horizontal lines mark the FOV of the MAX-DOAS instrument. **(b)** Plot of the camera data along the vertical plume cross sections marked in (a) as thin lines, their average as a thick red line and the MAX-DOAS elevation scan taken in the specified time frame as a solid blue line. The dotted blue lines mark the limits of the uncertainty interval of the MAX-DOAS elevation calibration.



of up to 1/2 FPS and $1920 \times 1200$ pixels. The instrument response to a wide range of target column densities, ranging up to
$1 \cdot 10^{18}$ molec cm$^{-2}$, has been examined in a numerical instrument model. A linear instrument response has been observed
within that range, making the instrument easy to calibrate. An examination of the signal-to-noise ratio has shown that the
ideal NO$_2$ column density in the gas cell of the instrument is approximately $4 \cdot 10^{18}$ molec cm$^{-2}$. Furthermore, under realistic
conditions, a detection limit of about $2 \cdot 10^{16}$ molec cm$^{-2}$ is expected. A study on the cross sensitivity to trace gasses other than
NO$_2$ was carried out for water vapour and O$_4$. Under assumption of realistic column densities of these species the magnitude
of the cross sensitivity of the instrument was predicted to be below an instrument signal equalling $-3.2 \cdot 10^{16}$ molec cm$^{-2}$
of NO$_2$. The predictions of the instrument model were verified in a proof-of-concept laboratory measurement, where four
test cells were filled with different concentrations of NO$_2$. Then their column densities were measured with a conventional
DOAS setup and the NO$_2$ camera. We noticed agreement between both instrumental setups within their uncertainties for all
test cells and between the camera results and the predictions of the instrument model. The average relative deviation between
model prediction and camera result amounted to 18.2%. We present the results of a field measurement at the coal power
plant Großkraftwerk Mannheim. The camera measured an average NO$_2$ plume SCD of $(4.74 \pm 2.00) \cdot 10^{16}$ molec cm$^{-2}$ and
an average plume diameter of $(78 \pm 34)$ m. By examination of an off-plume area the detection limit of this measurement was
estimated to be at $\Delta S = 1.89 \cdot 10^{16}$ molec cm$^{-2}$, however, the uncertainties of the evaluation procedure, mainly the background
estimation, increased the overall uncertainty to $\Delta S = 2.00 \cdot 10^{16}$ molec cm$^{-2}$. A mass flux analysis was carried out on the basis
of image sequences. For this purpose, the optical flow between pairs of consecutive images was estimated with a Farnebäck
algorithm, which yielded average horizontal wind speeds of $(0.94 \pm 0.33)$ m s$^{-1}$ and a resulting mean NO$_2$ mass flux of
$(7.41 \pm 4.23)$ kg h$^{-1}$ ($\widehat{=} (64.5 \pm 36.8)$ tons yr$^{-1}$).

Finally, the camera data were compared to elevation scans performed by a MAX-DOAS instrument. We found fair agreement
in magnitude and position of the plume signal, as measured by both instruments and identified the elevation calibration and
the poor spatio-temporal sampling of the MAX-DOAS in combination with the dynamic variability of the plume position as
the main uncertainty. Future measurements with these aspects in mind may yield more satisfying results. An improved optical
setup within the instrument could be considered. By including a beam splitter, the light for both sensor arrays could be collected
from a mutual lens, thus eliminating the need to correct for differences in the otherwise two lenses as a potential error source,
especially the cumbersome background fitting routine. Furthermore, this would make alignment of the cameras' optical axes
obsolete. In its current form the instrument is easily transportable and highly cost efficient with a build price of less than 2,000
Euro.

*Data availability.* All data is available from the authors upon request.

*Video supplement.* A series of camera images was assembled into a video sequence. It shows consecutive NO$_2$ camera images of the GKM
measurement from 08:53 to 09:05, where the observed NO$_2$ signal was especially strong. See Kuhn (2021).





*Author contributions.* LK, JK, TW and UP developed the question of research. LK and JK conducted the laboratory and field measurements. LK and JK developed the instrument model and the instrument prototype. LK characterized the instrument, evaluated the data, and wrote the manuscript, with all authors contributing by revising it within several iterations.

*Competing interests.* The authors declare that they have no conflict of interest.

*Acknowledgements.*





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
