# Peer review of "The NO2 camera based on Gas Correlation Spectroscopy"

_Atmospheric Measurement Techniques, 2021_

## Author Comment (AC1)

**Reply to RC1 by Emmanuel Dekemper**

The authors would like to thank Emmanuel Dekemper for the thorough review of their manuscript and the creative and helpful improvements. Below we reply to the raised issues one by one. Comments by Emmanuel Dekemper are printed in black, and the authors' replies in blue.

Please note, that during the review process, a new Figure and a new equation (Fig. 5 and eq. (15)) were added to the manuscript. All references to Figures or equations within this reply (except within direct quotes of sentences added to the revised version of the manuscript) refer to the **old** numbering, i.e. the one that the referees had seen when writing their reviews.

The manuscript presents a new instrument for the ground based remote sensing of atmospheric NO2. The sensitivity to the target species is provided by the principle of the gas correlation spectroscopy (GCS), but the main objective of this instrument is to become a quantitative imager of 2D fields of NO2. This NO2 imaging capability introduces this new instrumental concept in the very small family of imaging instruments capable of retrieving quantitative information on UV-VIS species (SO2 cameras, AOTF-based NO2 camera, and I-DOAS instruments to my knowledge). As such, this manuscript bears a strong original content, and can be considered as the seminal paper for a new class of NO2 imaging instruments.

The paper is well structured: it first explains the principles of the GCS, then presents a prototype and some results obtained in the laboratory, followed by the analysis of data acquired during an outdoor campaign at a coal-firing power plant. The quality of the text is very good: descriptions are exhaustive, the English is of very good level, and most figures are clear. The reader will not only find all the needed details to fully grasp the instrument principle, and what it can do, but he will also discover a honest account of real-life conditions performance, which is very much appreciated.

In the following, I'm listing a number of questions/remarks on scientific aspects. This review ends with the few typos/technical issues that I could spot.

**Specific comments/questions:**

My comments follow the structure of the paper:

- Introduction: The first part of the section lacks a few references about the harmfulness of NO2. Also, I believe that NO2 exposure limits recommended by the WHO have recently been lowered for NO2. Please check this and update if needed. Also, it might be interesting to list the previous remote sensing instruments which have employed the GCS method. I know about the satellite instrument HALOE, but others, even ground based might have existed.

The section has been updated accordingly. The statement regarding the health risks of $NO_2$ exposure now has two references. The WHO recommendation for annual exposure was changed to the most recent value (2021) of 10 $\mu$g m$^{-3}$.

Three other applications of GCS methods were already listed where GCS is introduced in the introduction: Ward and Zwick (1975); Drummond et al. (1995); Wu et al. (2018). Each of the references deals with a different practical application of the method. A reference to the HALOE instrument (Baker et al. (1986)) was added.

- Section 2.1: In the beginning of the section it could be wise to simply remind that the model assumes that the pressure and temperature dependence of the cross-section is neglected. Also, I appreciate, in eq.(4) that the authors didn't overlook the temporal variability by applying an integration over time. But as no parameter is exhibiting any temporal dependence, I think it is ok to replace the integral by a product with the exposure time.

A sentence explaining that the pressure and temperature dependence of the absorption cross sections were ignored throughout section 2 was added:

*"In the following, all absorption cross sections are considered constant, i.e. their slight dependence on pressure and temperature is neglected."*

The specified integral can indeed be written as a product, please refer to eq. (13).

- Section 2.1, line 123 and below: You are rightfully stressing the delicate alignment of the two optical channels. Actually, as your processing algorithm contains a normalization of one image by the other, the consequence is that the derived quantity gets more sensitive to parameters which might not be as identical as you expect. For instance, the pixel response non-uniformity (PRNU) map of the two sensors could be quite different, and therefore not get properly cancelled by the normalization. Therefore I think it is good to acknowledge the potential differences across the two channels by adding a subscript in eq.(7)-(8) and further. In particular, I would add the subscript "c" to \eta and \mu in eq.(8). With this notation, the reader will be in a better position to appreciate the attempt of canceling the optical effects with the normalization performed in eq.(12).

The correction procedure used by the authors ("flat field correction", or FFC) is a well-established method of correcting non-homogeneities in images, which works based on dividing the measurement image by a reference image taken against a radiometrically uniform background. This procedure also accounts for fixed pattern noise in general, including PRNU by gain normalisation. The authors have clarified this by text, adding the sentence

*"Even with entirely identical optical setups, the two camera sensors may have slightly different pixel response non-uniformity (PRNU) maps."*

in order to clarify, that PRNU is corrected by the FFC.

The authors have decided to omit the proposed subscript c to $\eta$ and $\mu$. The reason is that the FFC is assumed to correct $\eta$ and $\mu$ to the same values for both channels.

- Section 2.2, line 145: The assumed range of NO2 SCDs should be justified.

A sentence justifying the assumed range of $NO_2$ SCDs was added:

*"The assumed range of $NO_2$ column densities is justified as follows: In order to measure column densities much lower than $10^{16}$ molec $cm^{-2}$, the exposure time would need to be increased significantly, resulting in poor temporal resolution. At the same time, even strong $NO_2$ pollutions in the atmosphere typically do not exceed $10^{18}$ molec $cm^{-2}$, assuming realistic viewing geometries."*

- Section 2.2, line 161: From my own experience, relying on manufacturer data is risky. For quantitative measurements as the ones you intend to do, I would recommend to measure the detector parameters, and the optics transmittance by yourself, with the needed spectral resolution (which is very often very coarse in manufacturer's datasheet).

The authors agree in principle. However, the measurement method relies on narrowband spectral variations. Broadband transmissions, like that of the camera lens or the quantum efficiency mainly affect the SNR, but hardly the overall instrument response. Other instrument specific quantities (like the transmission spectrum of the bandpass filter) were validated by laboratory measurements and were found to agree with the vendor's information.

- Section 2.3: This section seems to be an attempt to get rid of the spectral integrals inherent to the method. In my opinion, the price to pay to achieve an analytical, integral-free expression for the effective optical thickness is too high. The reader can also be disturbed as the objective of this attempt is not fully clear. I would recommend to remove this section, in favor of stressing again that the measurements are not spectrally-resolved, and therefore a direct access to the target air mass SCD is not possible.

The purpose of this section is to present a purely analytical expression of the instrument response in a GCS-based measurement for arbitrary target gasses. The section is not highly relevant for the GCS-based $NO_2$ camera. However, for new GCS-based remote sensing methods it could be used for quick assessment of the instrument's capabilities because no knowledge of the spectral structure of the absorption cross section is required.

The authors have decided to move this section to the appendix (see Appendix A).

- Section 3, line 258: I was a bit surprised by the reported dark signals. It is quite strange to report values at temperatures well above the normal working conditions (e.g. 20°C). Also, the dark signal precision (24+/-9) is quite poor, much worse than the shot noise. This questions the accuracy of the dark current removal. But as the exposure times are orders of magnitude smaller than the second, I guess it is not really an issue. However, the text does not address the characterization of the ADC gain, which might be pixel-dependent, and detector-dependent. Perhaps providing the steps that you apply during the "L1" processing for the data would help the reader to understand exactly what is done.

The dark signal is listed at a temperature of 50 °C, because this is the average operating temperature of the two camera modules. This significant warm-up is due to the small form factor of the modules. The original sentence

*"A thermal dark signal of $(24 \pm 9)$ e$^-$ s$^{-1}$ at a sensor temperature of $50$ °C and a doubling temperature of $(6.1 \pm 0.1)$ °C were found"*

was changed to

*"A thermal dark signal of $(24 \pm 9)$ e$^-$ s$^{-1}$ at a sensor temperature of $50$ °C, which is approximately the average operating temperature of the camera modules due to their small form factor, and a doubling temperature of $(6.1 \pm 0.1)$ °C were found".*

Overall, the information on dark signal is included for the purpose of completeness and supposed to tell the reader, that a dark signal correction is obsolete. This is also stressed in the short example

*"(. . . ) the contribution of the dark signal to the total measured camera signal is negligibly small (e.g. below 0.05 % for an exposure time of 30 ms and a sensor saturation of 50 %)".*

The gain of the camera modules is set to a fixed value. As explained earlier, non-uniformities of the ADC gain are assumed to be accounted for by the flat field correction.

An explanation of the "L1" processing of the data was added by the authors:

*"From a technical perspective the retrieval of the camera data follows the typical pattern of digital imaging: Inside the camera modules, the incoming photons detach electrons from the semiconductor material of the camera chip (characterized by $\eta$). That charge is digitized (characterized by the fixed ADC gain K in units of [e$^-$ ph$^{-1}$]) and saved as 16-Bit grayscale image files.".*

- Section 4.1, eq.(25): This equation is not correct, as the ratio of the J-terms, followed by the log does not remove the spectral integrals contained in each J-term. At best, you obtain a biased estimate of the column. The importance of this bias can be assessed with a numerical experiment. Negligible or not, this aspect should be addressed.

This point was rightfully raised. Indeed, the term in eq. (25) is only an approximation of the column inside the gas cell of the instrument. A numerical experiment was run to assess the quality of this approximation. A new figure:

[Figure]

and an explanatory paragraph

*During measurements, $S_c$ must be determined so that $k^{-1}(S_c)$ can be computed. For this purpose, $S_c$ could be directly measured using a second instrumental setup, such as a DOAS instrument. However, in many measuring scenarios it is more practical to determine $S_c$ on the basis of the acquired images alone. For this purpose, an off-plume region of the imaged scene, where $S = 0$ is assumed, is used, and $S_c$ is approximated by*

$S_c = \ln\left(J/J_c\right)/\overline{\sigma}$

*where $\overline{\sigma} \approx 5.1 \cdot 10^{-19}$ cm$^2$ molec$^{-1}$ is the absorption cross section of $NO_2$, averaged over the spectral range from 430 to 445 nm. The validity of this approximation was verified numerically, as displayed in Fig. 5. For a cell column density of $S_c = 4 \cdot 10^{18}$ molec cm$^{-2}$ (this value will be reasoned in the following paragraph), the proposed approximation underestimates the true value of $S_c$ by less than $2 \cdot 10^{17}$ molec cm$^{-2}$.*

was added.

- Section 4.2.1: The vertical gradient that you observe like in figure 12 (a) is intriguing. You argue that it could come from the inhomogeneity of the reference measurements, but this does not really hold as the term J_ref(i,j)/J_c,ref(i,j) largely cancels broad variations of the sky signal across the FOV. While you have assessed the false signal induced by H2O, and O4, you haven't addressed the impact of not modelling the Rayleigh and aerosol extinctions. As the spectral integration is not removed in eq.(12), it is not excluded that the extinction by the Rayleigh scattering, given its wavelength-dependence (15% decrease over the filter bandwidth) is not completely canceled in the normalization. Given that the air mass increases as the pointing elevation decreases, this could perhaps explain the vertical gradient you are observing...

The relevance of such broadband extinction can be assessed as follows:

Rayleigh scattering is known to have a $\lambda^{-4}$ wavelength dependency. This would be the dominant modulation of the radiance spectrum if there were no further broadband extinction due to aerosols ("blue sky"). Realistically, the modulation due to broadband extinction can be assumed to modulate the radiance spectrum with some factor between $\lambda^{-4}$ and $\lambda^{-0}$. The influence of broadband extinction was estimated by modelling the two most extreme cases, i.e., computing the calibration curve of the instrument (like in Fig. 4) for (i) an unweighted radiance spectrum and (ii) a radiance spectrum weighted with $\lambda^{-4}$.

This study has shown that the contribution of the broadband extinction affects the instrument response by less than 1 % and is thus considered negligible.

Two explanatory paragraphs:

*Furthermore, eq. (12) points towards a crucial benefit of the proposed measurement principle. While other correlation methods for remote sensing typically operate with two channels in different spectral domains (e.g. an on- and an off band channel in filter spectroscopy based $SO_2$ cameras), the spectral domain of the two channels is identical in GCS. Additionally, that domain is typically restricted to a few dozen nanometers using a bandpass filter. This makes the instrument insensitive to broadband extinction, i.e. by Rayleigh scattering or due to aerosols, given that their extinction coefficients vary only very slightly throughout the spectral domain the instrument operates in. The instrument response to broadband extinction is examined numerically in sec. 2.2.*

and

*In addition to water vapour and $O_4$ the modelled instrument response to broadband extinction was investigated. Rayleigh scattering has a wavelength dependence of $\lambda^{-4}$, while extinction due to larger particles shows weaker wavelength dependence. It was verified in a numerical experiment, that the instrument response curves displayed in Fig. 4 vary by less than 0.5 % when the assumed irradiance spectra are scaled by $\lambda^{-4}$ and $\lambda^0 = 1$ respectively. This demonstrates that the $NO_2$ camera is practically insensitive to broadband extinction.*

The origin of these vertical gradients remains an open and important question, however, the background fitting method we propose seems to be able to correct them efficiently enough.

- Section 4.2.1: You are reporting a column of 2.72x10^18 in the cell, whereas it was explained that the best performance is achieved with a cell of 4x10^18. I also had understood that this was the value used in the

prototype. Why such a deviation with respect to your ideal case? Is it caused by using eq.(25)? Why didn't you estimate this value with the DOAS instrument?

A strong deviation of the column inside the cell was observed on multiple field trips, whenever the instrument was exposed to an environment that was usually much colder and/or brighter than the laboratory, where the prototype was built. The loss of $NO_2$ in the gas cell could be explained by the $NO_2 \leftrightarrow N_2O_4$ equilibrium. Additionally, the higher the irradiance, to which the gas cell is exposed, the higher the photolysis rate of the contained $NO_2$. It was verified after the field trip, that the $NO_2$ column inside the instrument's gas cell had recovered after some time in the laboratory. It was once planned to deliberately "overfill" the gas cell, so that it reaches the ideal column density when brought outside. However, this turned out to be very hard to achieve with the available $NO_2$ reservoirs, from which the gas cell was filled.

Measuring the column inside the cell requires complete disassembly of the instrument, which was considered unnecessary and too risky in the field, given that the column inside the cell can be computed with sufficient accuracy from the measurement images themselves. Furthermore, the high optical depth of $NO_2$ in the cell (on the order of unity) is not suitable for applying the DOAS technique in the blue spectral range.

- Section 4.2.1, figure 14: It is very difficult for the reader to clearly distinguish the plume from the background. Please consider a color scale which would help the reader to observe the abundance of NO2 in the plume.

A colormap was chosen to make the plume more pronounced. For consistency, the same colormap was applied to the corresponding figures of sec. 4.2.

- Section 4.2.2, 2nd paragraph: This paragraph spans 41 lines, and contains a very technical discussion. Please consider splitting in more paragraphs, or even better, divide in subsections. A subsection dedicated to your analysis of the influence of the choice of reference area would make sense for instance.

This section was divided by introducing a new subsection named "4.2.3 Uncertainty analysis".

- Section 4.2.5: The comparison with the MAX-DOAS measurements delivers results which are a bit deceiving, but it also seems like you attempted to compare data of very different temporal resolution. Can't you revisit your comparison in order to make sure that you have the highest temporal consistency? In other words, one would expect to have used a different set of camera images for comparison with each MAX-DOAS point. Working with a global average doesn't put you in the best conditions... In addition, you report an average plume signal of 1x10^16, which is below your detection limit. This is striking for the reader, and would deserve a little explanation.

Referee #2 has strongly urged to remove this section entirely, which the authors have agreed to. The reason a global average was used is that the plume signal is extremely weak compared to the detection limit of the instrument. Using a global average allows to just barely identify the plume. The approach suggested by Emmanuel Dekemper was originally attempted by the authors but delivered useless results.

**Technical comments**

- General comment about significant decimal: Everywhere in the text, values are reported with too many decimals compared to the uncertainty of the measurements. This starts already in the abstract, but affects the manuscript globally. This particularly concerns the columns in molec./cm^2, and also the mass fluxes.

The authors agree, that in the paper some results are given with too many decimal places. Those decimals have been reduced to a sensible number of digits, i.e. the instrument's spatial resolution is now given as 0.9 m × 0.9 m instead of 0.92 m × 0.92 m

This change was not applied to intermediate results, e.g. the contents of (and references to) Table 1 and Table 2, which are used for further computation and should remain untouched.

- Figures 5, 6, 7, 20: those figures have issues with labels or annotations.

This issue was fixed.

- L4: gasses -> gases

This was changed as suggested.

- L17: "momentary" ?

This word was considered obsolete and removed without replacement.

- L19: "...is highly portable and cost efficient at building..." -> "...is highly portable for building..."

This was changed as suggested.

- L28: add a comma after "time"

The comma was added as suggested.

- Figure 1: Are you sure that (a) is not the filled cell, and (b) the empty one? This is what I would have assumed based on the brownish color of cell (a)...

This observation is correct, and the caption of the figure was changed accordingly.

- L64: "Lamber-Beer" -> "Lambert-Beer"

This was changed as suggested.

- I found the unit "phe" quite uncommon. Why not using "e-" for representing photo-electrons?

The units were changed as suggested.

- L118: I would not speak of "spectral channels" when referring to the two optical channels of the NO2 camera.

The authors prefer to leave this unchanged. They acknowledge, that "spectral channel" may be an unconventional use of the word, however it bears a strong analogy to DOAS terminology. In DOAS, the SCD of the target trace gas is extracted from the information contained in the many spectral channels, by application of the Lambert-Beer law. We compute an SCD from two pieces of information, namely $J$ and $J_c$ (or four pieces, if the reference signals are considered), also by application of the Lambert-Beer law. What we call "spectral channel" is therefore the analogue to the "spectral channel" in DOAS.

The sentence:

*"We refer to the two measurements $J_{(i,j)}$ and $J_{c,(i,j)}$ as spectral channels in analogy to the widely used DOAS terminology."*

was added to make the intention of the authors clearer.

- L119: "... functions as a measure ..." -> I think it is more correct to say that it is a quantity exhibiting a monotonic sensitivity to S.

The authors prefer to leave this unchanged, given that "functions as a measure" portrays the purpose of computing $\tilde{\tau}$ more clearly to the reader.

- L143: add a comma after "For this"

The comma was added as suggested.

- L147: In don't think that t_exp has been defined before

$t_{\exp}$ is now defined under eq. (4), where it first appears.

- L151: double "of"

The obsolete "of" was removed.

- L365: remove the comma after "showed"

The obsolete comma was removed.

- L384: "two cameras inside of the instrument" -> "two cameras inside the instrument"

This was changed as suggested.

- L386: "These displacements manifest as strong..." -> "These displacements manifest themselves as strong..."

This was changed as suggested.

- L392: you introduce the subscript "i" for your columns, whereas I guess that everywhere before, "i" was a row index (like in matrix algebra convention)...

The row indices have been changed from $i$ to $j$, which would be the standard matrix algebra convention for a column index.

- L619: this value of 1/2 FPS is new. I had 1/12 in mind from the paper. Please check

In l. 15, the paper says 1/12 FPS, but in l. 619, the paper says indeed 1/2 FPS. The reason is, that the single images of the GKM measurement were recorded at a frame rate of 1/2 FPS, but in order to evaluate them, six consecutive images were averaged. This reduces the effective framerate to 1/12 FPS. However, the instrument is in principle able to record with frame rates higher than 1/12 FPS (again, here it was demonstrated, that 1/2 FPS is technically possible). An explanatory sentence

*"In order to increase the SNR of this measurement and smooth the plume signal, sequences of six images were averaged over, reducing the effective frame rate to 1/12 FPS and the resolution to $1350 \times 600$ pixels"*

was added.

- L623: Split the paragraph after "expected."

This was changed as suggested.

- L630: Split the paragraph after "18.2%."

This was changed as suggested.

---

## Author Comment (AC2)

**Reply to RC2 by Referee #2**

The authors would like to thank Referee #2 for the thorough review of their manuscript and the creative and helpful improvements. Below we reply to the raised issues one by one. Comments by referee #2 are printed in black, and the authors' replies in blue.

Please note, that during the review process, a new Figure and a new equation (Fig. 5 and eq. (15)) were added to the manuscript. All references to Figures or equations within this reply (except within direct quotes of sentences added to the revised version of the manuscript) refer to the **old** numbering, i.e. the one that the referees had seen when writing their reviews.

**General evaluation**

This manuscript introduces a novel measurement approach for the fast imaging of strong NO2 emission sources, such as stack plumes observed in power plants. The proposed NO2 camera is an application of the gas correlation spectroscopy, that was successfully used in the past e.g. for CO measurements in the infrared spectral range. The method is here extended to NO2 measurements in the visible spectral range. The study is largely exploratory in scope and concentrates on (1) a theoretical analysis of the measurement principle including estimates of the expected performances, (2) the verification of model predictions using a proof-of-concept instrument demonstrating the feasibility of the technique, (3) results from first measurements in the field at a large German power plant, and (4) comparisons with simultaneously recorded MAX-DOAS measurements. The proposed approach is very attractive since it is simple in concept and potentially inexpensive, which opens possibilities for future deployment at larger scale. In its current state, however, the system remains very experimental and a number of technical difficulties are still to be solved before such a camera can be ready for routine measurements in the field. Nevertheless I found the manuscript very interesting. The simple theoretical model is convincing and addresses several aspects of the measurements performances, such as sensitivity, selectivity, detection limit, etc. Model estimates are found to be in good agreement with actual measurements performed using the proof of concept instrument, which validates the approach. Also first measurements in the field show convincing results, for a stack plume of moderate strength. It also illustrates the main technical limitations of the current instrumental design. It also provides an interesting discussion on emission flux estimates performed using the camera, which arguably represents a promising application for future developments/applications. The last chapter on the comparison with MAX-DOAS measurements is however disappointing and somehow confusing. The authors struggle in a lengthy discussion to explain the potential reasons for a lack of agreement between both techniques, which result from a suboptimal operation of the DOAS system, a lack of time synchronisation and also calibration issues. In its current state, this comparison does not bring much to the study. I therefore strongly recommend to remove it and concentrate on an optimization of the experiment for a future publication. This reservation being made, I found the manuscript innovative, well written and definitely suitable for publication in AMT.

**Detailed comments**

Pg. 1, abstract: the first two sentences of the abstract could be omitted from the abstract. Such general information is generally provided in the introduction of the paper.

**The two first sentences of the abstract have been removed.**

Pg. 2, l. 54: I find the formulation "immanent asynchrony of the push-broom scheme" a little bit obscure in the present context. I suppose you mean that because of the need to scan in one spatial dimension, the information is recorded sequentially, which can lead to image deformation effects. Please confirm or clarify.

The interpretation of Referee #2 is correct. "immanent asynchrony of the push-broom scheme" means that when recording an image by scanning row by row, the individual rows composing the final image are asynchronous relative to each other.

The authors have changed the specified paragraph to:

Although modern hyperspectral cameras can reach adequate spatio-temporal resolution, some problems remain. Methods that rely on a push-broom scheme suffer from time delays between the rows (or columns) of the recorded images. Furthermore, spectrally resolving instruments are usually expensive and bulky. Pg. 3, l. 74: the instrument concept requires that the NO2 contained in one of the cells remains stable during measurements. Doesn't this requirement imply that the cell temperature must be stabilized to maintain the NO2/N2O4 ratio at a constant value? This question should maybe be addressed in the section dealing with instrument model calculations and uncertainties of the method.

This is a valid point and was considered by the authors. When setting up the experiments, the instrument was left running for a while (10 - 30 minutes) so that thermal equilibrium could settle inside the instrument case.

Remaining variations of  $S_c$  principally have an influence on the measurement, because the instrument calibration depends on  $S_c$ . However, the correct calibration factor  $k^{-1}(S_c)$  can be derived from  $S_c$  (see Fig. 4 for context), which again can be computed from each camera image, provided that an off-plume region with S = 0 is contained in it. For context, see eq. (25).

Following up to a comment in RC1, an additional explanatory paragraph regarding this procedure was placed in sec. 2.2:

During measurements,  $S_c$  must be determined so that  $k^{-1}(S_c)$  can be computed. For this purpose,  $S_c$  could be directly measured using a second instrumental setup, such as a DOAS instrument. However, in many measuring scenarios it is more practical to determine  $S_c$  on the basis of the acquired images alone. For this purpose, an off-plume region of the imaged scene, where S = 0 is assumed, is used, and  $S_c$  is approximated by

$$S_c = \ln \left( J/J_c \right) / \overline{\sigma}$$

where  $\overline{\sigma} \approx 5.1 \cdot 10^{-19} \text{ cm}^2 \text{ molec}^{-1}$  is the absorption cross section of NO2, averaged over the spectral range from 430 to 445 nm. The validity of this approximation was verified numerically, as displayed in Fig. 5. For a cell column density of  $S_c = 4 \cdot 10^{18}$  molec cm-2 (this value will be reasoned in the following paragraph), the proposed approximation underestimates the true value of  $S_c$  by less than  $2 \cdot 10^{17}$  molec cm-2.

along with a new Figure:

This should make it clear to the reader, that variations of  $S_c$  can be accounted for with sufficient accuracy.

Pg. 4, l. 93-96: are the units of radiances and irradiances important for this particular application? Certainly not for the measurement itself which is based on intensity ratios. But maybe this information is needed for the instrument model calculations. Please clarify whether absolute radiance values are used in this study.

Absolute radiances are used in the model calculations. In eq. (5), the radiance spectrum of the light source  $L_0$  is a radiance spectrum in units W nm-1 m-2 sr-1.  $L_0$  is also a spectrum of absolute radiance values. The reason for that is that for the determination of the SNR, e.g. in eq. (16) and (17), terms arise that no longer only depend on signal ratios, but absolute camera signals (which depend on absolute incoming radiances). For example, increasing the incoming radiance by a factor of 100 increases the SNR by a factor of 10.

Pg. 4, l. 100: I suppose that the wavelength dependence of the quantum efficiency indicated here is a property of the silicium-based detectors used for the measurements, which explains the limited spectral range (UV-Vis-NIR). Note that the use of the sun as a light source also limits the applicable spectral range.

**The authors agree, and the sentence**

"The wavelength dependence of  $\eta$  typically restricts the integration to the near ultra violet (UV), the visible, and near infrared regions of the electromagnetic spectrum."

**was changed to**

"The wavelength dependence of  $\eta$  and the spectrum of the light source (typically scattered sunlight) usually restrict the integration to the near ultra violet (UV), the visible, and near infrared regions of the electromagnetic spectrum."

**to reflect the Referee #2's comment.**

Pg. 13, l. 269: the fact that the adjustment of the alignment of the two cameras is scene-dependent represents a major limitation for operation in the field. Can you further develop the reason why this is the case? I understood from the last sentence of the conclusions that the use of another instrumental design could solve this issue. It would be nice to introduce this possibility with a bit more details in the main part of the manuscript.

When the optical axes of the two cameras are not aligned, they will diverge significantly at long distances. For example, if the optical axes are shifted by just  $0.1^{\circ}$  (0° would be perfectly parallel axes), an object in 2 km distance would appear shifted by 2 km · sin( $0.1^{\circ}$ )  $\approx 3.5$  m. Since eq. (12) computes the logarithmic ratio of the camera images, shifted structures are a source of strong false signals.

In theory, this problem could be solved once and for all by aligning the two axes perfectly and never touching the setup again. However, during transport of the instrument it is unavoidable, that the axes misalign slightly. Therefore, whenever the instrument arrives at the measurement site, the axes must be realigned.

Referee #2 has pointed out rightfully that the wording of this sentence is misleading and the corresponding sentence was changed from

"This adjustment is scene-dependent and of crucial importance in order to eliminate shifts in the FOVs of the two cameras"

 $\operatorname{to}$

"This adjustment is of crucial importance in order to eliminate shifts in the FOVs of the two cameras"

The alignment of the two cameras, only requires a few seconds using the thumb screws of the instrument and has not been regarded as a major limitation by the authors.

In l. 642, the idea of an instrument with a mutual optical setup for both channels and a beam splitter is briefly mentioned. Such an instrumental setup would have the potential to overcome shifts more easily, given that all light would be collected by a mutual lens.

Pg. 15, Table 1: Is there any particular reason why the uncertainty on the DOAS measurements of cell 2 so much larger than for other cells?

Please note that the uncertainty of cell 3 and cell 4 are larger than that of cell 2. The uncertainty is given by the uncertainty of the DOAS fit routine, but it may also vary from cell to cell, because the cells are of different size, have different glass thickness, etc.

Pg. 15, Fig. 9: this figure would gain being enlarged a little bit. Especially panel (a) is difficult to interpret.

**The figure was enlarged as suggested.**

Pg. 18, Fig. 12: again panels (a)(b) and (c) in this figure are very small and difficult to read. I suggest separating them from the two other panels and creating two separate figures. Since this figure shows the first illustration of an actual plume measurement with the camera, it deserves to be displayed in a more prominent way.

The panels (a), (b), and (c) of that Figure were enlarged but the authors decided to let them remain in a mutual Figure, given that panel (d) and (e) directly relate to them.

Pg. 19, l. 367: again these results demonstrate that the stability of the NO2 concentration in the reference cell is important, which suggests that an active stabilization of the temperature of the cell is needed (to constrain the NO2/N2O4) ratio.

This issue is resolved in the same manner as discussed in one of the previous points, regarding variations of  $S_c$ . When the measurement images are divided by the reference images taken against the blue sky (see eq. (12) for reference), and measurement image and reference image were recorded with different  $S_c$ , then a constant signal offset  $\tilde{\tau}_0$  appears on the resulting signal image. This is corrected as explained in the paper and happens on an image-by-image basis. The variations that Referee #2 has mentioned here are fully corrected for.

Pg. 22, Fig. 22 and related discussion in pg. 23: the need to manually define the mask used to estimate the background 'out-of-plume' signal is also an important limiting factor for the technique. Do you see a possibility to overcome this difficulty either through an instrumental modification or by means of a more elaborated processing technique? If yes, it would be interesting to further discuss this question, maybe in a short section dedicated to perspectives for improvement of the technique.

This is a valid concern.

One solution would be to always use the "Full-FOV mask", see Fig. 16 (i). This eliminates a manual selection, but produces worse results. As Table 2 shows, the "Full-FOV mask" approach underestimates the NO2 SCD by approximately 25 %.

Another solution would be some form of automated plume detection. However, finding a general algorithm for such purposes is a very hard exercise. The authors would like to refrain from theorising much about this, but a short explanation was added to underline the importance of the plume mask to the whole evaluation:

"In the future, more elaborate methods for the separation of plume and background should be investigated. Generally, this would be achieved by image segmentation, for which a variety of methods exists. However, finding an ideal method that generalizes to other plume shapes and viewing geometries would require a study on its own."

Pg. 25, l. 465: at the end of the sentence, refer to section 4.2.4 where the question of the NO2/NOx ratio is explicitly analysed.

A sentence was appended, referring to the specified section:

"The  $NO_2/NO_x$  ratio of the plume is further investigated in sec. 4.2.5."

Pg. 28, section 4.2.5: as already pointed out in my general comments, I strongly recommend to remove this section from the paper. My feeling is that it brings confusion and does not help consolidating the measurements obtained with the camera. I would suggest replacing it by a small section outlining the possible improvements that can be envisaged for the instrument and eventually the data evaluation.

The authors have decided to remove the specified section.

Furthermore, the final part of sec. 5 now includes a brief listing of further ideas for future improvements:

"In the future, the following improvements to the instrument should be implemented: Firstly, the optical setup inside the instrument can be further optimized. By including a beam splitter, the light for both sensor arrays could be collected from a mutual lens, thus eliminating the need to correct for differences in the otherwise two lenses as a potential error source, especially the cumbersome background fitting routine described in sec. 4.2.1. Additionally, there exist camera modules with much lower read-out time than the ones used in our prototype, increasing the overall photon budget available for measurements. Secondly, the instrument would benefit from thermal stabilization in order to maintain a more stable NO2 column inside its gas cell. This way, the evaluation procedure would rely less on successfully determining  $S_c$  (see sec. 2.2) and  $\tilde{\tau}_0$  (see sec. 4.2.1) from an off-plume region of the camera images. Thirdly, when measuring NO2 emissions from a strong source as in sec. 4.2, the evaluation routine could be made significantly less ambiguous by implementing an automated image segmentation algorithm to separate the plume and off-plume regions of the individual images."

Pg. 34, l. 1: in fact, if I understand correctly, the camera was operated at a reduced resolution of 1300 x 600 pixels (accounting for the windowing applied to reduce the read-out time).

Correct, the camera was operated on a reduced resolution of  $1350 \times 600$  pixels (equalling an extent of 1245 m width and 551 m height, as seen in Fig. 14). A sentence was added a few paragraphs later, where the limitations in framerate and resolution specific to the GKM measurement are listed:

"In order to increase the SNR of this measurement and smooth the plume signal, sequences of six images were averaged over, reducing the effective frame rate to 1/12 FPS and the resolution to  $1350 \times 600$  pixels."

Pg. 34, 623: '... a detection limit of about 2e16 molec/cm2 is expected...'. Here I would add that this was confirmed by measurements using the proof-of-concept instrument.

The sentence was changed from

"Furthermore, under realistic conditions, a detection limit of about  $2 \cdot 10^{16}$  molec cm-2 is expected."

 $\operatorname{to}$

"Furthermore, under realistic conditions, a detection limit of about  $2 \cdot 10^{16}$  molec cm-2 is expected, which was later confirmed using the instrument prototype."

**Spelling, typos**

Pg. 2, l. 44: remove 'either'

The word "either" was removed.

Pg. 6, l. 128: change 'In reality this latter condition need not be perfectly filled' by 'In reality this latter condition does not need to be perfectly filled'

This was changed as suggested, but the word "filled" was exchanged by "fulfilled".

Pg. 7, l. 151: 'of' is duplicated between 'choice' and 'particular'

The obsolete "of" was removed.

Pg. 10, Fig. 5: there seems to be a confusion of the 'S' and 'Sc' notations in this figure. To my understanding, the x-axis of panel (a) should be labelled as 'Sc' as well as the legend of panel (b). Please check and adjust as needed.

The Figures are labelled as intended by the authors. It also seems that the panels are already labelled as suggested by Referee #2: The x-axis of panel (a) is labelled as  $S_c$ , as well as the entries in the legend of panel (b).

Pg. 10, Fig. 6: the species of which cross-sections are disp^layed do not show up properly in the legend where the applied scaling factors are given.

This issue was fixed.

Pg. 13, l. 258: I suppose that the sensor temperature is fixed at -50°C, and not +50°C as indicated here.

The temperature is not fixed at all. The overall temperature of the camera sensors depends on the heat produced by the units themselves and the ambient temperature, to which the cameras are exposed. Due to their small form factor, the camera modules indeed heat up to a temperature of around +50 °C. However, exposure times are so short, that dark signal can be neglected, even at such high temperatures (see also our replies to RC1).

Pg. 22, Fig. 16, end of first line: replace 'The left two' by 'the two left'

This was changed as suggested.